# Transfer learning with graph neural networks for improved molecular property prediction in the multi-fidelity setting

David Buterez [1] ✉, Jon Paul Janet [2], Steven J. Kiddle[3], Dino Oglic[4] & Pietro Lió [1]

We investigate the potential of graph neural networks for transfer learning and improving molecular property prediction on sparse and expensive to acquire high-fidelity data by leveraging low-fidelity measurements as an inexpensive proxy for a targeted property of interest. This problem arises in discovery processes that rely on screening funnels for trading off the overall costs against throughput and accuracy. Typically, individual stages in these processes are loosely connected and each one generates data at different scale and fidelity. We consider this setup holistically and demonstrate empirically that existing transfer learning techniques for graph neural networks are generally unable to harness the information from multi-fidelity cascades. Here, we propose several effective transfer learning strategies and study them in transductive and inductive settings. Our analysis involves a collection of more than 28 million unique experimental protein-ligand interactions across 37 targets from drug discovery by high-throughput screening and 12 quantum properties from the dataset QMugs. The results indicate that transfer learning can improve the performance on sparse tasks by up to eight times while using an order of magnitude less high-fidelity training data. Moreover, the proposed methods consistently outperform existing transfer learning strategies for graph-structured data on drug discovery and quantum mechanics datasets.

We investigate the potential of graph neural networks (GNNs) for transfer learning and improved molecular property prediction in the context of funnels or screening cascades characteristic of drug discovery and/or molecular design. GNNs have emerged as a powerful and widely-used class of algorithms for molecular property prediction thanks to their natural ability to learn from molecular structures represented as atoms and bonds[1–3], as well as in the life sciences in general[4–6]. However, their potential for transfer learning is yet to be established. The screening cascade refers to a multi-stage approach where one starts with cheap and relatively noisy methods (high-throughput screening, molecular mechanics calculations, etc.) that

allow for screening a large number of molecules. This is followed by increasingly accurate and more expensive evaluations that come with much lower throughput, up to the experimental characterisation of compounds. Individual stages or tiers in the screening funnel are, thus, used to make a reduction of the search space and focus the evaluation of more expensive properties on the promising regions. In this way, the funnel maintains a careful trade-off between the scale, cost, and accuracy. The progression from one tier to another is typically done manually by selecting subsets of molecules from the library screened at the previous stage or via a surrogate model that focuses the screening budget of the next step on the part of the chemical space

[1]Department of Computer Science and Technology, University of Cambridge, Cambridge, UK. [2]Molecular AI, BioPharmaceuticals R&D, AstraZeneca, Gothenburg, Sweden. [3]Data Science & Advanced Analytics, Data Science & AI, R&D, AstraZeneca, Cambridge, UK. [4]Centre for AI, BioPharmaceuticals R&D, AstraZeneca, Cambridge, UK. ✉e-mail: db804@cam.ac.uk

around the potential hits. Such surrogate models are typically built using the data originating from a single tier and, thus, without leveraging measurements of different fidelity.

For efficient use of experimental resources, it is beneficial to have good predictive models operating on sparse datasets and guiding the high-fidelity evaluations relative to properties of interest. The latter is the most expensive part of the funnel and to efficiently support it, we consider it in a transfer learning setting designed to leverage low-fidelity observations to improve the effectiveness of predictive models on sparse and high-fidelity experimental data. In drug discovery applications of this setup, low-fidelity measurements can be seen as ground truth values that have been corrupted by noise, experimental or reading artefacts, or are simply performed using less precise but cheaper experiments. For quantum mechanics simulations, low-fidelity data typically corresponds to approximations or truncations of more complex and computationally-expensive calculations, such that the low- and high-fidelity labels are closely related. Thus, it is natural to expect that incorporating low-fidelity measurements as an input feature into a high-fidelity model typically improves the performance on sparse tasks relative to predictors learnt using the high-fidelity data alone. Despite its apparent simplicity, even in the transductive learning setting (i.e., low-fidelity and high-fidelity labels are available for all data points), it is not trivial to define an adequate workflow that jointly uses both low- and high-fidelity labels. For instance, devising an end-to-end training scheme with low- and high-fidelity labels as part of the same model can be challenging for drug discovery applications, where the disparity between the numbers of respective observations is larger than two orders-of magnitude (e.g., the number of high-fidelity observations can be over 500 times lower than that of the low-fidelity ones). Previous work has successfully applied multi-fidelity learning on several problems[7], but as we show in "Comparison with the multi-fidelity state embedding algorithm" section that approach is unfortunately not effective in drug discovery. While successfully exploiting multi-fidelity data in the transductive setting is valuable on its own, virtually all high-throughput screening steps in drug discovery are followed by experiments generating high-fidelity measurements for molecules that were not part of the original screening cascade, i.e., lacking low-fidelity labels. Devising an in silico model of the low-fidelity portion of the screening cascade is thus highly desirable, as it enables the generation of low-fidelity representations (e.g., labels) for arbitrary molecules that were not part of the original funnel. This inductive learning capability is crucial, as drug discovery requires making predictions about molecules that have not been made yet, and therefore models that rely on transductive information (i.e., measured low-fidelity labels) are generally inapplicable. Even the highest throughput assays require that the molecules of interest must be synthesised first.

The main motivation for our transfer learning approach is the desire to leverage representations learnt from low-fidelity data to improve the predictive performance on sparse high-fidelity tasks. To support this goal, we incorporate modern deep learning architectures, more specifically graph neural networks, in our workflow. We consider two modes of transferring that knowledge into the high-fidelity models: (i) learning models for each fidelity independently, with the caveat that the high-fidelity molecular representation includes the feature(s) generated, as outputs, by the low-fidelity model(s), and (ii) pre-training a graph neural network on the low-fidelity data and then devising an effective fine-tuning strategy for the high-fidelity data. Both approaches naturally support transductive and inductive learning settings. While both modes of information propagation between different levels of fidelity are applicable to vanilla graph neural network architectures, it is not necessarily the case that representation transfer will yield performance improvements. More specifically, we have observed that transfer learning with graph neural networks has been underutilised

and have therefore performed a detailed empirical study to assess the capabilities of existing graph neural network architectures for transfer learning between observations associated with different levels of fidelity. Our empirical results indicate a critical shortcoming in standard architectures that severely limits their transfer learning potential on several learning tasks, namely the readout function responsible for aggregating embeddings of individual atoms into molecule-level representations. The design of the readout functions is a fundamental aspect of geometric deep learning, and a transition to neural network-based operators, also called adaptive readouts or neural readouts, over simple and fixed functions such as sum or mean has only recently been studied extensively[8]. More specifically, we leverage the famous attention mechanism[9], which is increasingly used in the life sciences in different forms[10–13]. However, adaptive readouts and their fine-tuning have not been characterised in relation to their transfer learning potential, which we address as part of this study. The transfer learning capabilities enabled by adaptive readouts are further supported with a supervised variational graph autoencoder designed to learn a structured and expressive chemical latent space that can be used for downstream sparse high-fidelity tasks.

The proposed framework has been evaluated on a heterogeneous collection of transfer learning tasks, including a drug discovery collection of 37 different protein targets (more than 28 million unique experimental protein-ligand interactions in total) and 12 quantum properties from the dataset QMugs (around 650K drug-like molecules). The analysis involves several different baselines, ranging from graph neural networks to random forests and support vector machines. Our empirical results highlight the importance of transfer learning in low-data regimes, which are encountered in drug discovery projects relying on high-throughput screening and are not uncommon in quantum simulations. More specifically, we vary the size of the training set on high-fidelity data and show that transfer learning can improve the accuracy of predictive models by up to eight times while using an order of magnitude less high-fidelity training data. In the transductive setting, we notice that the inclusion of the actual low-fidelity label typically amounts to performance improvements between 20% and 60%, and severalfold in the best cases. However, out of the total of 51 transductive experiments, transfer learning via label augmentation was the best-performing method in only 10 instances, with the graph neural network schemes introduced here proving as the most effective 80% of the time (Table 1). For the more challenging inductive learning setup, we notice substantial improvements in performance due to latter methods, typically between 20% and 40% in the mean absolute error and up to 100% in $R^2$. These performance improvements are mainly due to alleviating the shortcomings of non-adaptive readouts in classical graph neural networks. While on quantum mechanics problems standard graph neural networks are competitive, particularly if they are extensive and non-local (which agrees with the existing literature), they are still outperformed by fine-tuning strategies involving adaptive readouts. In drug discovery tasks, on the other hand, the standard/vanilla graph neural networks significantly underperform the baselines. For completeness, we also provide comparisons with state-of-the-art strategies for multi-fidelity learning and transfer learning with graph-structured data, in the form of the multi-fidelity state embedding (MFSE) algorithm proposed in[7] and a variation of pre-training and fine-tuning devised by[14]. Unfortunately, neither of the two approaches performs particularly well on drug discovery tasks. We take several steps towards ensuring that our framework is general by firstly extending to a scenario with real-world data that has more than two levels of fidelity. Furthermore, for this setting we also evaluate a model operating directly on 3D molecular structures (the SchNet architecture[15,16]) in place of a typical graph neural network, thus validating transfer learning with adaptive readouts on this family of widely-used and state-of-the-art models as well.

**Table 1 | A count of the best augmentation strategy for each of the three groups of datasets**

| | Label | Neural | | | | Sum | | | |
|---|---|---|---|---|---|---|---|---|---|
| | | Emb. | Pred. label | Hybrid label | Tune readout | Emb. | Pred. label | Hybrid label | Tune network |
| AZ | 2 | 3 | 1 | 2 | 8 | 0 | 0 | 0 | 0 |
| PubChem | 4 | 2 | 1 | 0 | 15 | 0 | 1 | 0 | 0 |
| QMugs | 4 | 5 | 0 | 0 | 3 | 0 | 0 | 0 | 0 |
| Subtotal | 10 | 10 | 2 | 2 | 26 | 0 | 1 | 0 | 0 |
| Total | 10 | 40 (Neural) | | | | 1 (Sum) | | | |

AstraZeneca (AZ), PubChem, and the QMugs 10K diverse set, in the transductive setting, and as ranked by '% MAE decrease' (see "Results"). The first row of headers (**Neural** and **Sum**) refers to the type of graph readout function used for the low-fidelity models, while the second row specifies the transfer learning strategy. The 10 cases where the labels are preferable are discussed in detail in Supplementary Notes 15 and Supplementary Table 4. Bold is used for the table headers.

## Related work

In terms of related work, introducing a funnel of increasingly expensive and accurate measurements is a common feature in molecular design. In drug discovery, this is exemplified by high-throughput screening (HTS) and follow-up assays, while in quantum mechanics (QM) there are different levels of theory. In all cases, practitioners are most interested in results from the most accurate level, both in transductive and inductive settings. High-throughput screening is one of the core methods for identifying starting points in drug development and it is responsible for approximately one-third of newly discovered drug candidates[17–19]. During HTS, the activity of millions of compounds against a target is evaluated using a multi-stage/tier approach, generating large data collections with different levels of fidelity[20]. The first stage corresponds to primary screening and consists of low-fidelity measurements for up to two million diverse compounds[21]. Primary screening traditionally acts only as a filter for identifying the most likely candidates for the more expensive and precise second stage, a confirmatory screening. The confirmatory step typically provides high-fidelity measurements for no more than 10,000 carefully-selected compounds, acting as another filter prior to lead optimisation. The confirmatory screening is typically done sequentially via multiple batches, whereas the primary screening is performed only once and it is impractical to run additional experiments at this scale as subsequent designs are typically synthesised in small batches. Another example where the funnel approach is relevant is the prediction of molecular properties by computational chemistry, typically with quantum mechanics simulations. In these cases, the accuracy of the simulations is tuned by using different levels of theory with different computational costs. Currently, QM simulations at the scale of millions of molecular geometries are possible at the density-functional levels of theory (DFT) or less accurate semi-empirical methods[22–24], and through approximations of the gold standard CCSD(T) method[25]. Due to extreme computational costs, the use of accurate methods such as G4MP2, CCSD(T), or CCSDT is only feasible for a few thousand molecular conformations with typically less than a dozen atoms[22,26].

In quantum chemistry, a family of established methods is given by the so-called Δ-predictors[14,23,27,28], which are designed for the transductive setting and focus on learning additive corrections of the low-fidelity measurements when estimating the properties of interest. These approaches are defined only for quantum properties and thus on a chemical space of limited diversity (e.g., only 3,114 unique molecules for ANI-1x/ANI-1ccx, with the heaviest atom being oxygen). Furthermore, not many high-quality datasets exists to support multi-fidelity data modelling, with the very recently introduced QMugs dataset[23] and our own MF-PCBA collection[29] being notable exceptions. As such, the standard choice for data, techniques, and model architectures remains single-fidelity, even for recent quantum machine learning efforts (e.g., Alchemy[30], QM7-X[31], nablaDFT[24]). Thus, the benefits of obtaining and holistically modelling molecular data produced at different fidelity levels remains insufficiently explored. In drug discovery, for instance, the benefits of leveraging millions of experimentally-derived molecular measurements at different fidelity levels is particularly understudied. As recent large-scale drug discovery efforts focus mainly on increased automation, reproducibility, and cost-effectiveness[21], the burden of making sense of the immense collections of molecular interactions falls onto computational methods. Integrative modelling of multi-fidelity data and improved molecule-level predictions could lead to cost-savings by reducing or avoiding some of the expensive wet-lab experiments, the identification of new, promising compounds, more informed experiments for ongoing projects, and hybrid wet-lab and in silico workflows. However, existing studies of real-world drug discovery data are built on a single-fidelity paradigm, even more so than in the case of quantum mechanics funnels[32–37].

Screening cascades with multi-fidelity outputs have a long-standing history in materials chemistry as well. For instance, Yang et al.[38] have recently proposed a two-step approach to improve the predictions of the short circuit density and fill-factor by leveraging data from low-fidelity simulators. The first step in that approach relies on an unsupervised autoencoder to extract a compressed representation of microstructure images. The second step then leverages the learnt latent space embedding augmented with a low-fidelity label to train a surrogate model on the high-fidelity dataset. In contrast to our approach, transfer learning is done on images and not on graph-structured data, and the latent space embedding is learnt without supervision done via low-fidelity data points. Furthermore, a completely unsupervised approach is unlikely to translate to molecules, as such representations are difficult to use for property-based downstream prediction tasks[39]. Another interesting approach to multi-fidelity learning is the composite neural network by Meng and Karniadakis[40] that aims to learn the parameters of inverse PDE (partial differential equation) problems with non-linearities. However, the approach is designed for problems where the correlation between low- and high-fidelity data is unknown, a case that is not generally encountered in drug discovery or quantum simulations. It has been applied in the context of PDEs with shallow and small neural networks and has only been evaluated on relatively small datasets (between 30,000 and 45,000 samples) with mainly uni-dimensional functions.

Multi-fidelity learning also has applications in active learning and Bayesian optimisation, where one aims to iteratively optimise a black-box function using a fixed budget of high-fidelity evaluations. The main motivation for this setup is the desire to leverage low-fidelity simulations to eliminate regions with low function values using an inexpensive proxy for a targeted property of interest. The expensive high-fidelity evaluations are then used in small but promising regions to quickly find the optima. Fare et al.[41] have used a multi-task Gaussian process in this setting for materials design and screening of molecules. While Gaussian processes have the advantage of being able to perform uncertainty quantification, the computational complexity (cubic) can

be a challenge for large-scale low-fidelity datasets. The problem setting in that approach is also significantly different from the one studied here, as their main focus is to extend Bayesian optimisation for materials design from a single-task sample-efficient setup to multi-fidelity models capable of leveraging information from different sources. Previously, Patra et al.[42] have also employed multi-fidelity Gaussian processes to predict polymer band gaps on a small dataset of 382 polymers. Perhaps the closest to our work is the custom message-passing algorithm by Chen et al.[7], which is the first notable work on multi-fidelity graph neural networks. The algorithm, which we refer to as multi-fidelity state embedding or MFSE for short, proposes a small modification to the standard message passing workflow, where in addition to node (atom) and edge (bond) messages and update functions, a state message is constructed and updated alongside them. The state embedding is a global attribute of the graph (molecule), and it is updated according to a fidelity encoding that is present for each data point. In other words, during training each molecule is associated with its structure, regression label, and a fidelity indicator (e.g., low or high) that is used to propagate fidelity-specific messages. The approach has two shortcomings: (i) as in Fare et al., jointly training on all fidelities can be problematic when the low-fidelity data outnumbers the high-fidelity samples by more than two orders-of magnitude, as the high-fidelity information can effectively be lost, and it remains to be explored if the state embedding mechanism can account for this (see "Comparison with the multi-fidelity state embedding algorithm" section), and (ii) coupling low- and high-fidelity training means that to enable high-fidelity inference, the model must be trained on the entirety of the low-fidelity data, dramatically increasing the training times and resource utilisation. In[43] a general approach has been proposed for recursively modelling more than two fidelity levels by means of feature augmentation where a high-fidelity data representation includes a predicted low-fidelity label associated with the previous screening step. This approach is similar to our baseline for the inductive setup where a low-fidelity proxy is obtained using a surrogate model.

## Results

This section provides a comprehensive empirical analysis of all the transfer learning techniques described in the "Feature augmentation via low-fidelity simulations" section ("Methods"). We evaluate the effectiveness of different feature augmentations for transfer learning given by: (1) explicit low-fidelity labels, (2) labels predicted by the low-fidelity models (with both sum and adaptive readout variants), (3) explicit low-fidelity labels during training and predicted ones during inference (a hybrid approach), (4) latent space embeddings generated by low-fidelity models (with both sum and adaptive readout variants) and two fine-tuning strategies tailored for graph neural networks: (5) pre-training and fine-tuning a standard GNN similarly to Smith et al.[14], and (6) fine-tuning of the adaptive readout in models that were pre-trained on low-fidelity data, while keeping fixed the weights in other layers. All the methods are compared relative to baselines trained exclusively on sparse high-fidelity data as the goal is to improve their predictive ability in that setting. The effectiveness of different strategies is illustrated on a suite of experiments with the following structure and goals:

1. Learning with standard and adaptive readout-based VGAEs exclusively on the low-fidelity data (Fig. 1, step 2, and Fig. 2, bottom) that is available in large, diverse, and heterogeneous datasets to assert their capacity to learn structured concepts ("Learning with graph neural networks via standard and adaptive readouts" section). This is a foundational contribution that indicates the potential of adaptive readouts for fitting the data and their ability to transfer knowledge from latent embeddings to new tasks via fine-tuning operations.

2. Learning predictive models on sparse high-fidelity data that incorporate the raw low-fidelity labels, indicating their utility for transfer learning in the transductive setting (see "Effectiveness of transfer learning strategies in the transductive setting" section).

3. Learning predictive models on sparse high-fidelity data that incorporate representations (e.g., latent space embeddings or predicted labels) generated by low-fidelity models in transductive (see "Effectiveness of transfer learning strategies in the transductive setting" section) and inductive (see "Effectiveness of transfer learning strategies in the inductive setting" section) settings.

4. Fine-tuning of low-fidelity models on high-fidelity data, highlighting a strategy where only the adaptive readout is retrained on small-sample tasks. Our empirical results demonstrate superb performance of this strategy and that it outperforms fine-tuning standard GNNs (see "Effectiveness of transfer learning strategies in the transductive setting" section).

5. Evaluating all the presented strategies while varying the high-fidelity training set size, demonstrating severalfold improvements while using an order of magnitude less training data (see "Effectiveness of transfer learning strategies while varying the size of the training sample in sparse high-fidelity tasks" section).

6. Evaluating the proposed strategies relative to established multi-fidelity and transfer learning techniques on a set of representative datasets. In addition to pre-training and fine-tuning standard GNNs[14], we also compare to the multi-fidelity state embedding (MFSE) approach of[7] and show that neither of them offers significant improvements on drug discovery tasks (see "Comparison with the multi-fidelity state embedding algorithm" section).

7. Evaluating an extension of transfer learning strategies to more than two fidelities. More specifically, we demonstrate that multiple lower-level fidelity inputs can be successfully integrated and that they can work synergistically, improving performance of downstream models when used jointly compared to individually incorporating them. In this suite of experiments, we use SchNet as the underlying architecture, further validating the effectiveness of transfer learning strategies across a different and widely-used family of GNNs (see "Extending to multiple fidelities" section).

We conclude this overview of our empirical study by highlighting two of the main reasons behind our extensive suite of evaluated augmentations (corresponding to points (1)–(4) above): (i) there are multiple possible ways of transferring knowledge from separately trained low-fidelity models and some might be more adequate for certain protein targets or quantum properties, dataset sizes, or fidelity correlations; (ii) every time an adaptive (neural) readout is used, we also report the result for the equivalent model with a sum readout (in an attempt to simulate an approximately similar inductive bias in the feature extraction part of graph neural networks). The proposed augmentations are agnostic to the underlying architectures and, thus, the main goal of these baselines is to illustrate the limited potential of standard GNN readout functions for transfer learning in multi-fidelity settings.

### Learning with graph neural networks via standard and adaptive readouts

To assess the potential of adaptive readouts for enabling high-capacity hypothesis spaces, we train two models on low-fidelity data, with identical architectures and hyperparameters but different readouts. More specifically, we use the sum operator as a representative of the standard readouts (performs similarly to mean and maximum on bioaffinity tasks[8]) and the Set Transformer as an adaptive (neural) readout[8]. As a further quality check for the drug discovery tasks, we consider a 'null hypothesis' that predicts the dataset mean for each

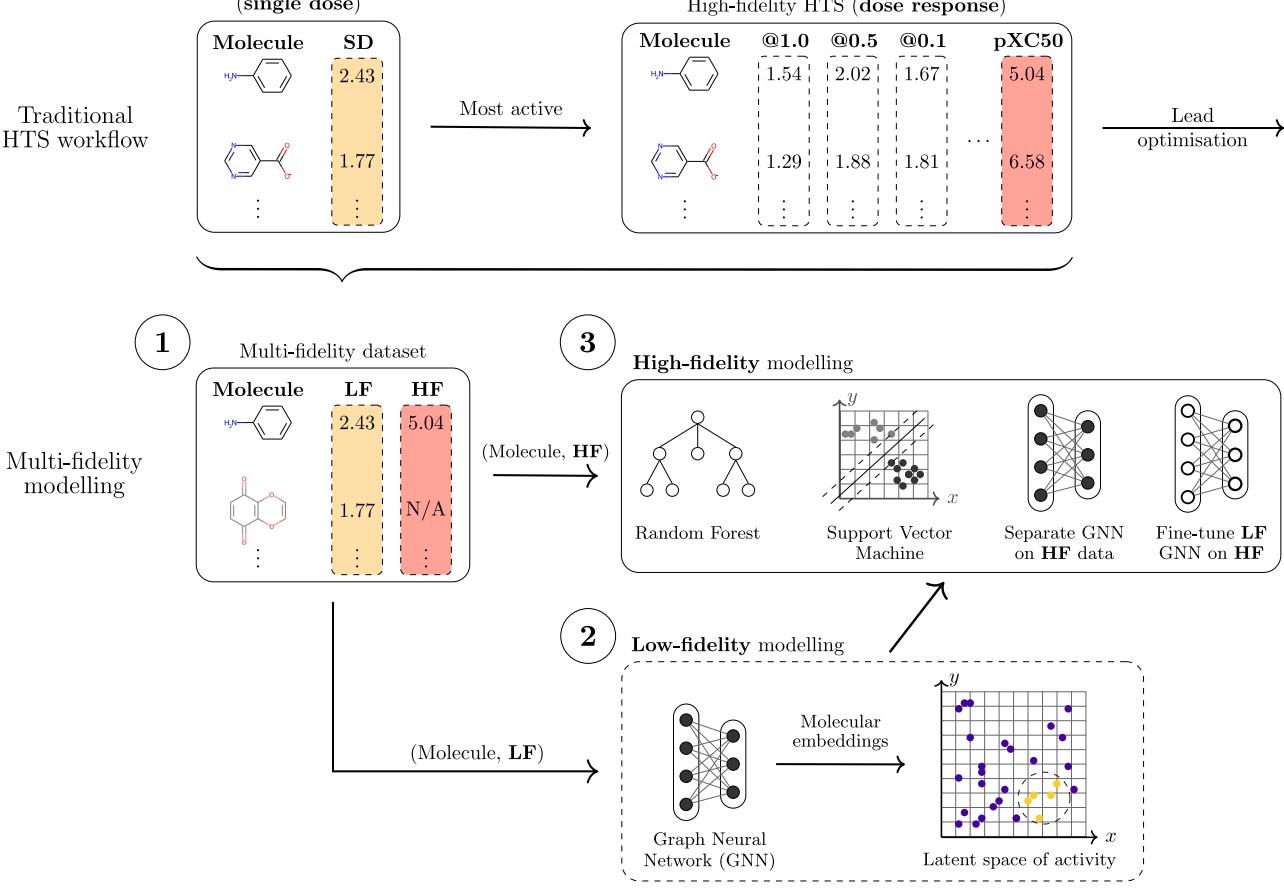

**Fig. 1 | An overview of transfer learning approaches that leverage low-fidelity data to improve predictions on sparse high-fidelity tasks.** The illustration depicts a typical drug discovery scenario where a large dataset of noisy observations is followed by a sparse, high-fidelity set of measurements obtained via expensive and time-consuming assays. The workflow (steps 1, 2, 3) is general and can be applied to other settings, e.g., quantum mechanics. (Top panel) A traditional HTS (high-throughput screening) experiment generates a massive but noisy set of low-fidelity measurements (primary screening, measuring the activity of molecules at a single concentration, indicated using the format '@ concentration'). An orders-of-magnitude smaller set of molecules is selected for a high-fidelity experiment, such as a confirmatory screen. (Middle panel, bottom) Our proposed transfer learning framework illustrated with three high-level steps in a drug discovery context. Firstly, the corresponding low-fidelity single dose (SD) and high-fidelity dose-response (DR) data are assembled into a multi-fidelity dataset. DR values correspond to confirmatory screens measuring a 'pXC50' activity value (the effect X ∈ {I = inhibitory, E = effective, A = activatory}), representing the concentration required for a 50% effect. DR is only available for a fraction of the entire dataset, hence some compounds are not available ('N/A'). Secondly, a graph neural network (GNN) is trained on the large primary screening dataset (**low-fidelity** or **LF**), modelling an extensive and diverse chemical space of interest. Finally, at the third step, the molecular structure, supplemented with the molecular fingerprint/representation learnt at the second step, is used to train models of confirmatory activity (**high-fidelity** or **HF**), including both deep learning and classical algorithms.

molecule, which is a relevant baseline because HTS data is biased toward low activity. We trained supervised VGAEs independently on all of the low-fidelity data for each multi-fidelity dataset. Our empirical results indicate that the performance of the sum readout models generally closely matches or even does worse than the null hypothesis (Fig. 3A, B) for all drug discovery tasks, but is more competitive on the quantum tasks, particularly for the extensive and global properties such as the total energy. This agrees with the existing literature and highlights the unique challenge posed by the HTS data. For drug discovery, the adaptive readout models offer severalfold improvements in the capacity to learn from the low-fidelity data, as measured by the MAE (Fig. 3) and $R^2$ (Supplementary Fig. 1), with smaller but noticeable improvements for QMugs. The uplifts observed here translate to better downstream performance in both transductive and inductive settings, as can be seen from "Effectiveness of transfer learning strategies in the transductive setting" section onward. At the same time, models using adaptive readouts exhibit a characteristic structuring effect on the latent space with respect to the low-fidelity domain, as visualised using 3D UMAP (Fig. 3C). For the drug discovery data, we notice a clear and often continuous demarcation of active and inactive compounds, an effect that is not present when using standard readouts, which generate scattered and less informative representations. Similar effects can be observed for quantum properties.

## Effectiveness of transfer learning strategies in the transductive setting

In this section, we systematically evaluate the performance of all the proposed low-fidelity augmentations on our HTS-based drug discovery collection and on QMugs in the transductive setting (i.e., real, experimentally-derived low-fidelity labels are available for drug discovery, with semi-empirical xTB labels for QMugs), with a standard train, validation, and test evaluation workflow (Fig. 4). The transductive setting acts not only as an excellent machine learning benchmark, but is also of practical use in drug discovery as it enables higher quality predictions in the high-fidelity domain for the millions of compounds that are present in the screening cascade. To this end, we train models on sparse high-fidelity data and incorporate different representations of low-fidelity information (strategy 1, and strategies 2–4 with both

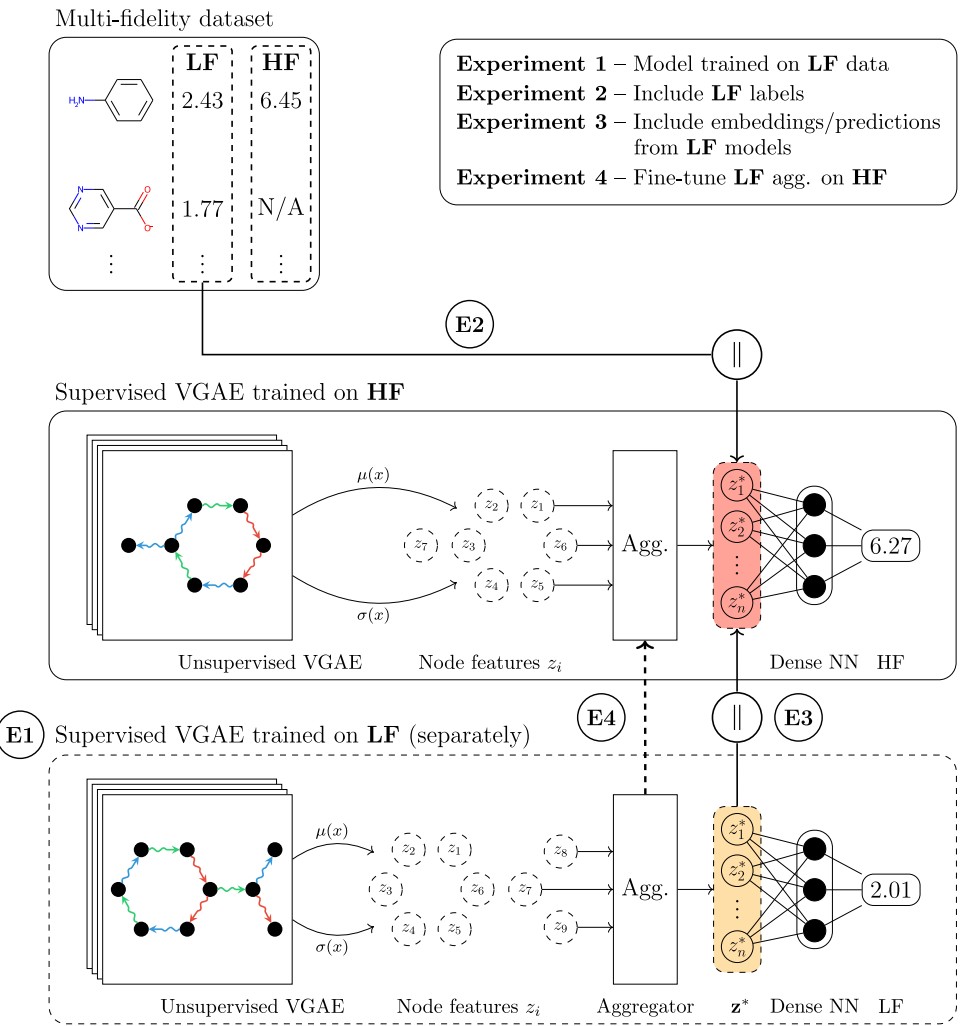

**Fig. 2 | The proposed supervised variational graph autoencoder (VGAE) presented schematically in a typical drug discovery scenario with a large and low-fidelity (LF) high-throughput screening dataset, and a sparse and high-fidelity (HF) confirmatory screening dataset.** Graph convolutions are used to propagate and learn atom-wise representations according to the connectivity imposed by the bonds, which are then aggregated into a single molecule-level representation or embedding (a fixed-dimension vector). The readouts are standard pooling functions, e.g., sum, mean, max, or neural networks (adaptive aggregators). The symbol ‖ denotes concatenation, $\mu(x)$ and $\sigma(x)$ denote the mean and standard deviation learnt by the VGAE, and 'Dense NN' is a multi-layer perceptron. The four workflows presented in this figure are listed in the top right and correspond to the first four experiments presented in "Results". A **low-fidelity** model is first trained with supervised information to produce latent space embeddings **z\*(E1)**. A separate model with the same architecture can then be trained to predict **high-fidelity** values, by concatenation with either the actual labels **(E2)** or embeddings/predictions generated by the **LF** model **(E3)**. A strategy unique to graph neural networks with adaptive readouts is that of pre-training a model on **LF** data as in **(E1)** and then fine-tuning exclusively the adaptive readouts with the VGAE layers being frozen **(E4)**. We also emphasise that the learnt low-fidelity embeddings/predictions can be integrated into any machine learning algorithm (e.g., support vector machines, random forests, etc.).

sum and neural readout variations), or fine-tune the low-fidelity models from "Learning with graph neural networks via standard and adaptive readouts" section (strategies 5 and 6). The number of times each strategy ends up as the best performing one is provided in Table 1.

For the drug discovery tasks, we noticed consistent decreases in MAE between 10% and 40% just by the inclusion of raw low-fidelity single-dose labels, with only a few exceptions (Supplementary Figs. 2 and 5–7 for the remaining datasets). Moreover, we notice that augmenting with neural embeddings can outperform the actual low-fidelity labels by up to 10%. The predictions generated by neural readout low-fidelity models, either by themselves or in a hybrid setting, are generally slightly worse than the embeddings but comparable to the raw labels (Fig. 4). In contrast, the embeddings and predictions produced by sum readout low-fidelity models struggle to achieve any decrease in MAE, occasionally even degrading the performance. The

inability of the sum readout low-fidelity models to fully model the data is particularly emphasised by the hybrid augmentation, where the actual experimentally-derived labels are used during training and the predicted ones for evaluation. The significant decrease in the effectiveness observed in this case indicates the disparity between the space covered by the raw labels and the information carried by sum readout low-fidelity models, which agrees with previous observations ("Learning with graph neural networks via standard and adaptive readouts" section).

In addition, we evaluate fine-tuning strategies for sum and neural readout low-fidelity models. The former, without a learnable readout function and any frozen components, generally struggles to improve to the same degree or is even worse compared to other sum-based augmentations. In contrast, pre-training and fixing the graph layers, followed by fine-tuning of the neural readout generally matches the other neural-based augmentations and is even preferable by a large

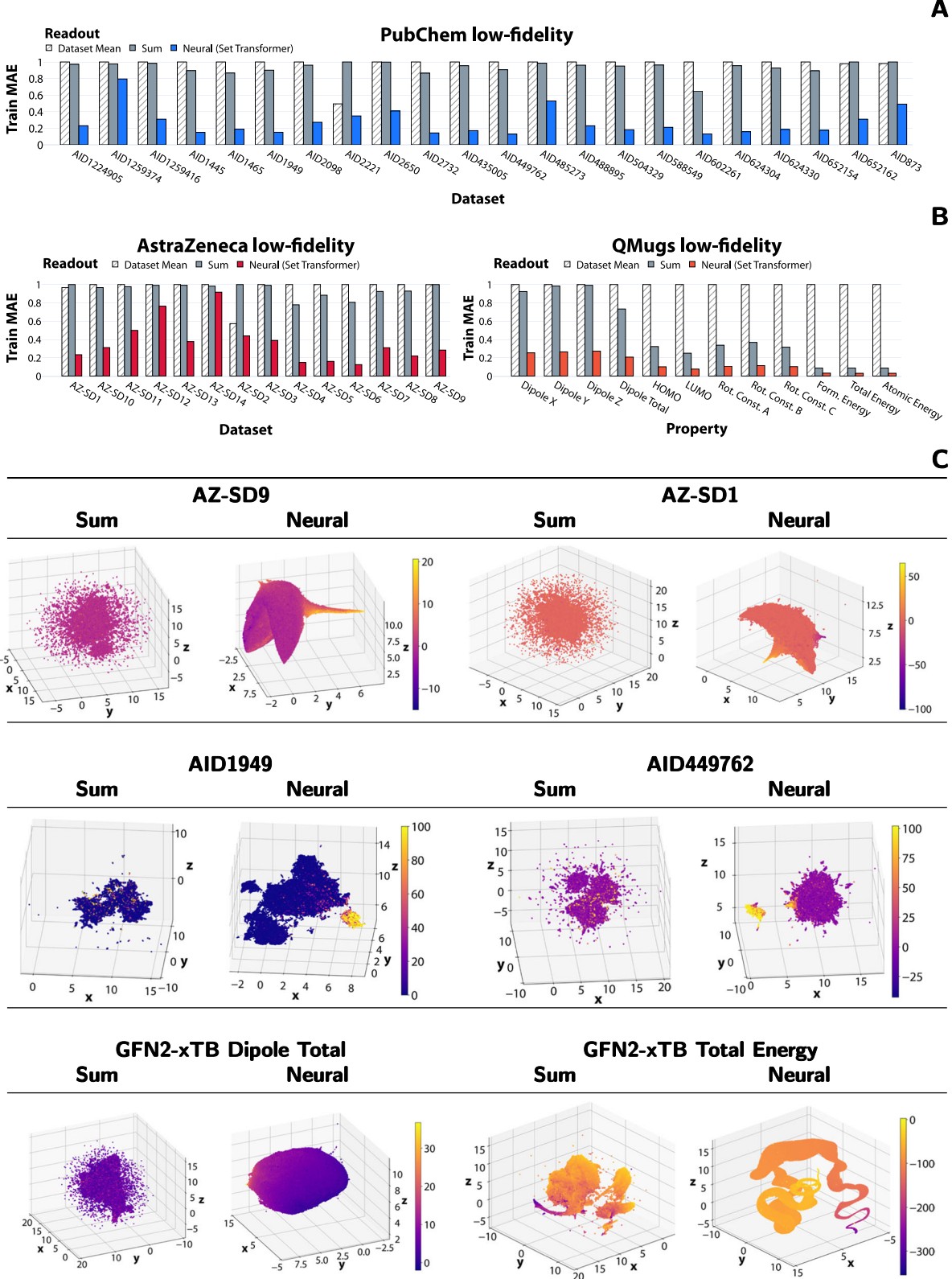

**Fig. 3 | Systematic evaluation of the ability of models based on different graph readout functions to learn from large-scale and complex datasets. A** Train MAE for the PubChem low-fidelity models, with a model predicting the dataset mean for comparison. The MAE values are scaled to the range [0, 1] for each dataset. **B** The same but for the AstraZeneca and QMugs models. **C** 3D UMAP latent space visualisations for a selection of low-fidelity models using sum and neural readouts.

Similar effects are observed for other datasets. The dataset sizes are: 1581928 (AZ-SD9), 1700745 (AZ-SD-1), 98472 (AID1949), 311910 (AID449762), and 647794 (QMugs after filtering). MAE denotes the mean absolute error and UMAP stands for uniform manifold approximation and projection. Source data are provided as a Source Data file.

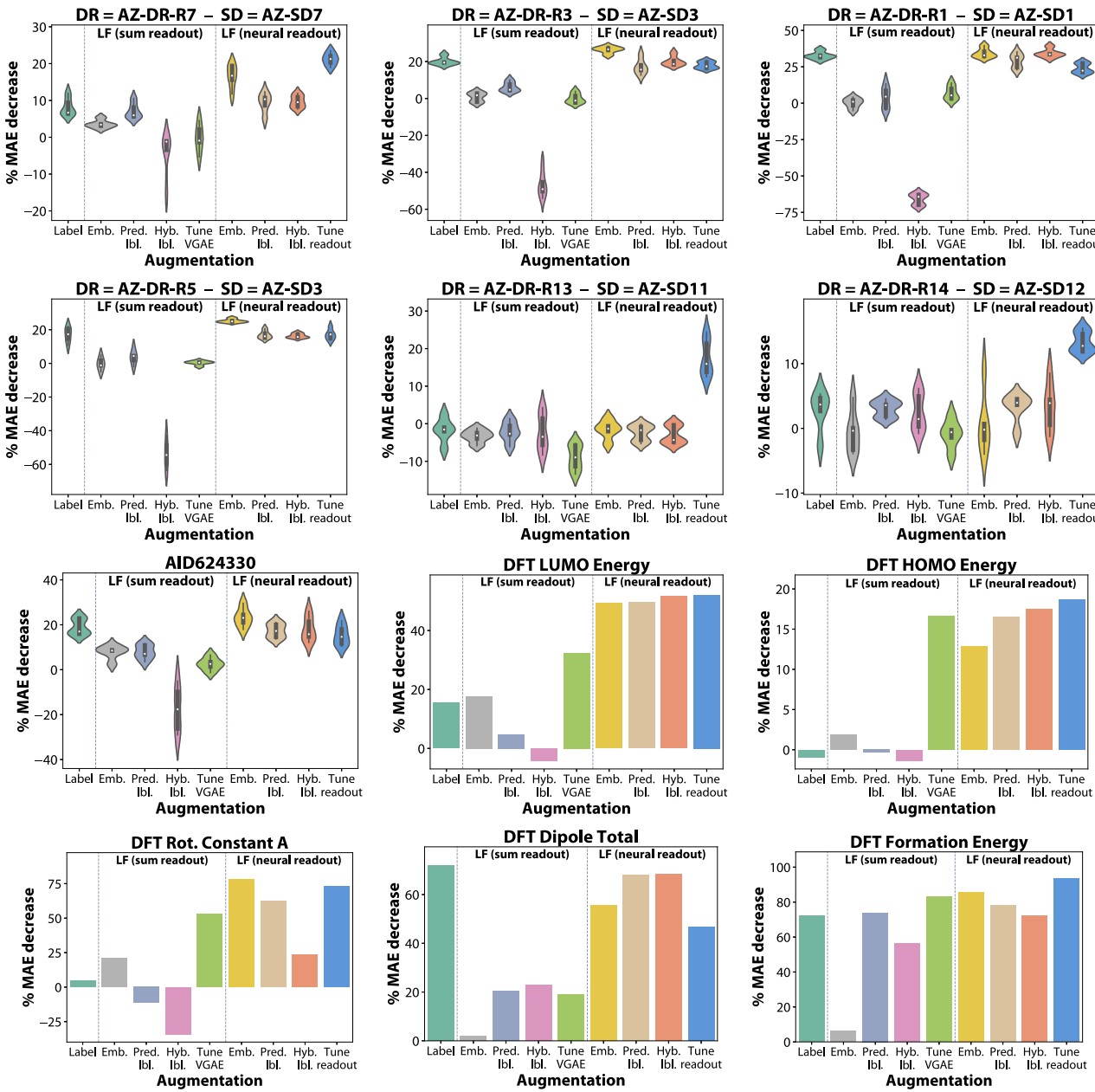

**Fig. 4 | Systematic evaluation of drug discovery (AstraZeneca, PubChem) and quantum mechanics (QMugs) datasets in the transductive setting.** The test set performance of high-fidelity models with augmentations based on sum and neural readout-based low-fidelity ('LF') models, including fine-tuning (denoted by 'Tune', see "Methods" section) is presented. 'Emb.' corresponds to the incorporation of low-fidelity embeddings, 'Pred. lbl.' corresponds to the predicted labels (outputs) of LF models, 'Hyb. lbl.' signifies training with raw labels and evaluating on LF-predicted labels, and 'readout' denotes the graph readout function. The AstraZeneca datasets are named based on the high-fidelity (DR, dose-response) and low-fidelity (SD, single dose) datasets. The abbreviations are: AZ AstraZeneca, AID assay identifier, VGAE variational graph autoencoder, MAE mean absolute error, DFT density-functional theory. The remaining results (other datasets) are available in Supplementary Figs. 2 and 5 to 7, with random forest and support vector machine results in Supplementary Figs. 3, 4 and 8 to 11. Source data are provided as a Source Data file.

margin for datasets such as AZ-DR-R7, AZ-DR-R13, and AZ-DR-R14 (Fig. 4). To mimic the HTS setting on QMugs, we selected a diverse and challenging set of 10K molecules (Supplementary Notes 5.1). The QM models are of particular interest as certain quantum properties are naturally additive and non-local, such that the sum readout is theoretically capable of modelling those tasks[44,45]. While we explore this dynamic in[46], here we report similar trends as for drug discovery, namely that the adaptive embeddings and fine-tuning the adaptive readouts outperform the raw labels by a large margin, with the exception of the four dipole properties which are particularly difficult to learn (see Supplementary Notes 15 for a possible explanation).

However, for a large selection of properties such as the HOMO and LUMO energies and rotational constants, the low-fidelity xTB labels are completely ineffective, whereas the adaptive readout strategies lead to improvements between 20% and 80% in MAE, depending on the properties. For additive properties such as the atomic, formation, and total energies, the sum readout is competitive, as expected, although it is still outperformed by adaptive readouts. In this transductive setting, for drug discovery and QMugs, we report a statistically significant relationship between the uplift in performance ('% MAE decrease') and linear correlation (Pearson's r) between the low and high-fidelity measurements (Fig. 5A).

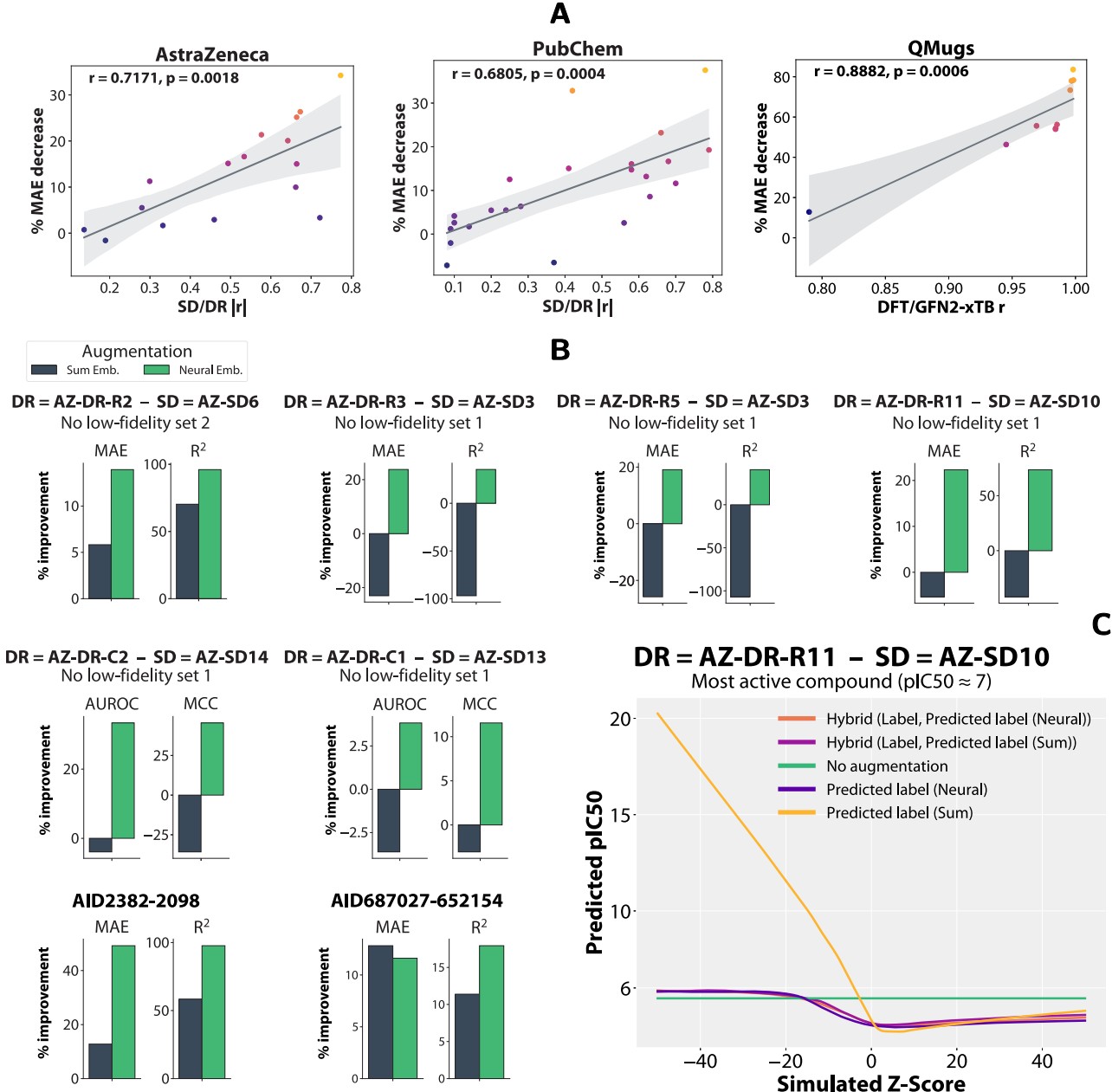

**Fig. 5 | Systematic evaluation of drug discovery (AstraZeneca, PubChem) datasets in the inductive setting and their learnt trends, and an analysis of dataset correlations. A** Scatter plots of the low-fidelity/high-fidelity correlation measured by Pearson's r for each dataset (x-axis) and the relative MAE decrease computed for the neural embeddings-augmented transductive models with regards to the non-augmented baseline (y-axis), with the regression line for the two variables and 95% confidence intervals for the regression. **B** Systematic evaluation of high-fidelity models using sum and neural embeddings in an inductive setting, where the previous train, validation, and test splits from Fig. 4 are used for training and testing is performed on compounds that were measured in subsequent HTS stages. We have also observed cases where both sum and neural-based augmentations did not provide uplifts or even decreased performance (Supplementary Fig. 12 for all the remaining datasets). **C** Example of a model evaluated in this

inductive setting where we supply low-fidelity labels (Z-Score) ranging from −50 to 50 in 0.5 increments. Models that rely on sum-based low-fidelity predictions learn nonsensical relationships with linearly increasing pIC50 values. As expected, this is alleviated by the hybrid augmentation, where training uses the raw labels. In contrast, neural-based predictions are initially more conservative than the baseline and slowly surpass the non-augmented models in terms of predicted activity. The multi-fidelity drug discovery datasets are named based on the high-fidelity (DR dose-response) and low-fidelity (SD single dose) datasets. The abbreviations are: AZ AstraZeneca, AID assay identifier, DFT density-functional theory, GFN2-xTB geometry frequency noncovalent eXtended tight binding, pIC50 negative logarithm of the half maximal inhibitory concentration, MAE mean absolute error, $R^2$ coefficient of determination, AUROC area under the receiver operating characteristic, MCC Matthews correlation coefficient. Source data are provided as a Source Data file.

## Effectiveness of transfer learning strategies in the inductive setting

We now turn our attention to one of the most challenging scenarios encountered during computationally-assisted early-stage drug discovery: improving predictions in an inductive setting with possible

out-of-distribution test samples relative to the training data. Here, we used the entire high-fidelity data discussed in the previous section for training, and used test molecules that were selected later in the drug discovery campaign (i.e., lacking low-fidelity labels) for evaluation. This set of test molecules is referred to as a 'no low-fidelity set'. It is

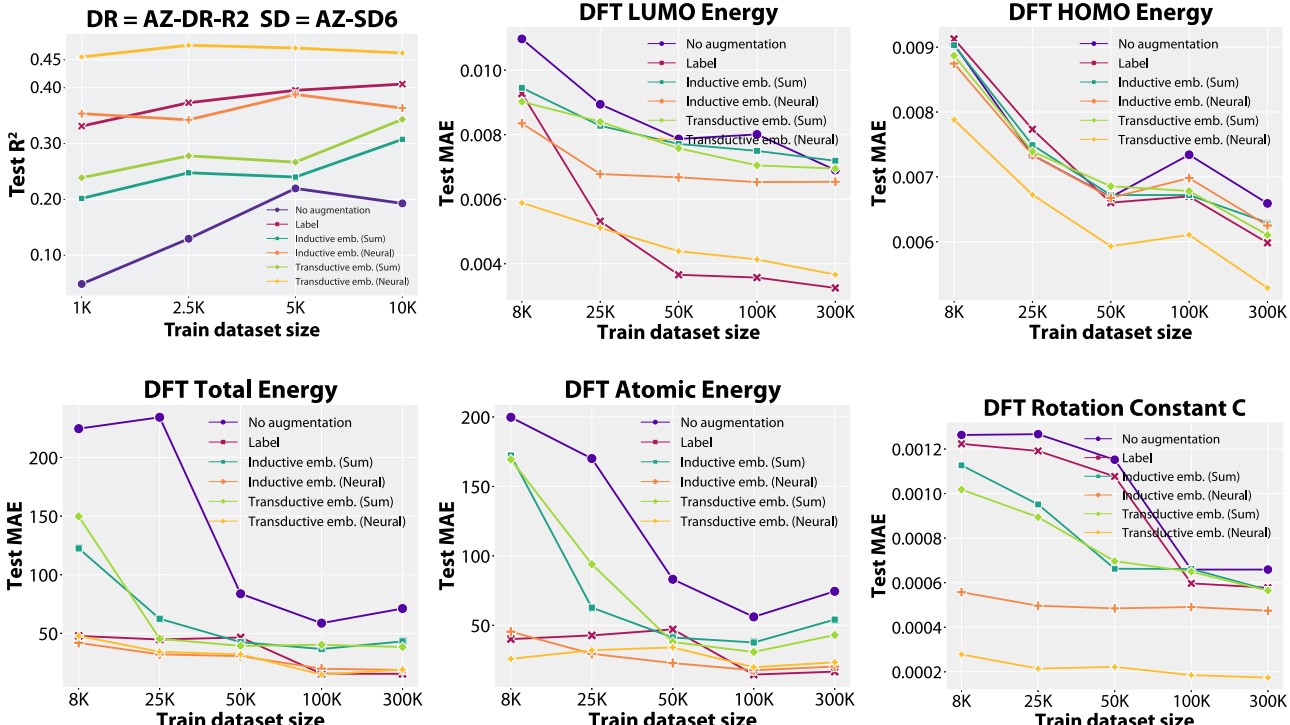

**Fig. 6 | Evaluation of transfer learning models with different training set sizes.** Test metrics for high-fidelity models with fixed validation and test sets but varying train set sizes and different strategies, in transductive and inductive settings. The rest of the quantum properties are available in Supplementary Fig. 13. The abbreviations are: AZ AstraZeneca, DFT density-functional theory, HOMO highest occupied molecular orbital, LUMO lowest unoccupied molecular orbital, MAE mean absolute error, $R^2$ coefficient of determination. Source data are provided as a Source Data file.

possible to encounter multiple such sets within the same project. For brevity, we focus only on embeddings generated by models with adaptive readouts and the sum counterpart for comparison.

For all categories of datasets (originating from public or private sources; regression and/or classification tasks), we report substantial increases in performance when using adaptive readouts, often between 20%–40% in MAE and up to 100% in $R^2$ (Fig. 5B), with a remarkable uplift for the classification task AZ-DR-C2, where the MCC increased from 0.69 to its maximal value of 1.0. We are again able to identify that the embeddings produced by sum readout low-fidelity models lead to performance degradation compared to the plain non-augmented baseline. While the augmentations given by the sum readout models carry some low-fidelity information, their effectiveness is further limited by a different problem. Namely, for AZ-DR-R11 we selected the most active compound in the test set (Fig. 5C) and supplied the high-fidelity models with simulated low-fidelity labels (i.e., Z-Score values ranging from −50 to 50 in 0.5 increments). We leveraged models augmented with predicted labels as we can directly provide artificial values, unlike for the variation with embeddings. We discover that models that rely on sum readout-based low-fidelity models fail to learn a meaningful relationship between the provided label and the target pIC50, as the pIC50 value steeply increases to unlikely quantities even for common Z-scores in the range of −20 to −30.

**Effectiveness of transfer learning strategies while varying the size of the training sets in sparse high-fidelity tasks**

To illustrate the importance of leveraging low-fidelity measurements in small-sample regimes characteristic of high-fidelity tasks, we evaluate the previously discussed transfer learning strategies while varying the training set sizes of high-fidelity data and maintaining fixed validation and test sets in the same high-fidelity domain. We selected the largest drug discovery dataset (AZ-DR-R2), having slightly under

12,000 molecules in the high-fidelity domain (confirmatory screen) and assembled random training subsets of sizes 1K, 2.5K, 5K, and 10K, such that larger training samples contain all the molecules of the smaller subsets, with fixed validation and test sets of equal sizes from the remaining data outside the 10K training set. For QMugs, we used the same diverse set of 10K molecules with high-fidelity DFT labels as a starting point and challenging train, validation, and test splits of size 8K, 1K, and 1K respectively (Supplementary Notes 5.1). We then generated high-fidelity training subsets of sizes 25K, 50K, 100K, and 300K following the same strategy as for drug discovery, keeping the validation and test sets of size 1K fixed. We demonstrate the effects of transfer learning in both transductive (low-fidelity datasets contain the molecules appearing in the high-fidelity subsets) and inductive settings (adjusted low-fidelity datasets where the molecules appearing in high-fidelity subsets are removed). For brevity, we illustrate only the transfer by embeddings strategy, generated by low-fidelity models trained in either transductive or inductive settings.

The feature augmentations based on adaptive and sum readout low-fidelity models are compared to baseline non-augmented models and the raw low-fidelity labels, with test set metrics reported in Fig. 6. On AZ-DR-R2, we notice a large uplift just by the simple inclusion of low-fidelity labels (from 0.049 to 0.331 in $R^2$ for the 1K split). In contrast, the best performance of non-augmented models is achieved for the 5K training set size ($R^2$ of 0.220), which is more than 50% lower than the performance on the 1K set with low-fidelity labels. Adaptive embeddings produced in a transductive setting improve over 8 times over the baseline $R^2$ (0.455) and by 37.39% over the raw labels. The adaptive embeddings produced in an inductive setting generally match the raw labels, even outperforming them for the 1K training set size. Both sum embeddings perform similarly, improving upon the baseline but not matching the raw labels. We notice similar trends for different quantum properties. For example, in a transductive setting, the MAE is almost halved for LUMO energy, and it is reduced by 4−8

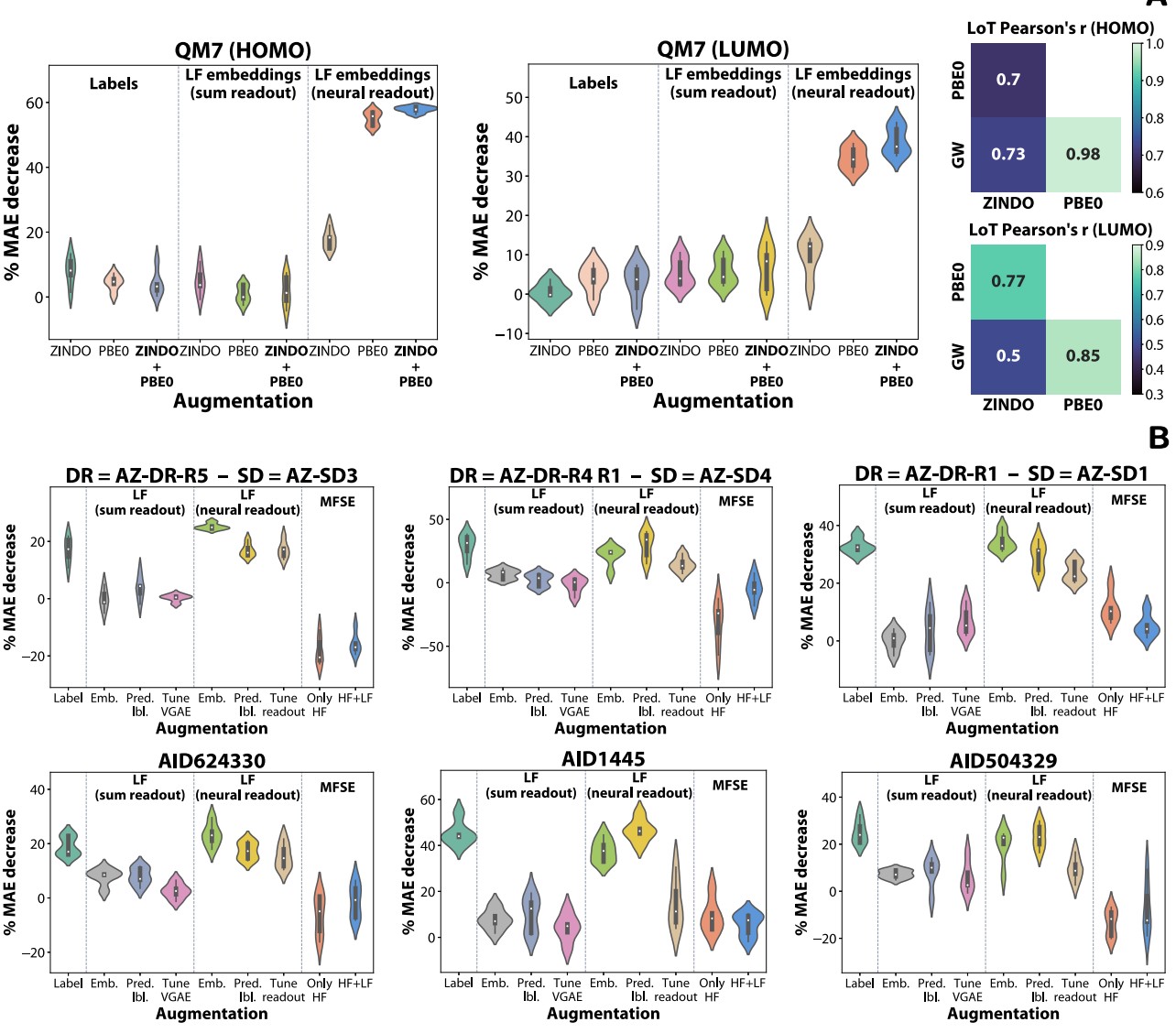

**Fig. 7 | Evaluation of transfer learning models with multiple fidelities and a comparison with the multi-fidelity state embedding algorithm. A** Test metrics for QM7b models leveraging three fidelities corresponding to the ZINDO, PBE0, and GW levels of theory (`LoT') and their correlations. **B**. Evaluation of the transfer learning strategies in a transductive setting and in the context of the established multi-fidelity state embedding (MFSE) method. The multi-fidelity drug discovery datasets are named based on the high-fidelity (DR dose-response) and low-fidelity (SD single dose) datasets. The abbreviations are: AZ AstraZeneca, AID assay identifier, HOMO highest occupied molecular orbital, LUMO lowest unoccupied molecular orbital, MAE mean absolute error. Source data are provided as a Source Data file.

times on the total and atomic energies and the rotation constants for the most relevant 8K training split. Interestingly, for that split the embeddings learnt in the significantly more challenging inductive setting often outperform the actual low-fidelity label. Depending on the quantum property, these observations can generalise even to the splits that use more high-fidelity training data (e.g., for total and atomic energies, rotational constants, and LUMO and HOMO energies to an extent). Representations learnt by sum readout low-fidelity models can lead to an improvement compared to the baseline models (no augmentation), however by a much smaller amount.

### Comparison with the multi-fidelity state embedding algorithm

In this section, we compare the proposed strategies with a recently proposed graph neural network architecture devised specifically for multi-fidelity learning on molecules, which we refer to as 'multi-fidelity state embedding' or MFSE for short[7]. We evaluate MFSE on a

representative selection of 3 public and 3 AstraZeneca drug discovery datasets. The datasets with the largest performance uplift when using low-fidelity information were chosen to maximise the chance of MFSE being effective. Unfortunately, MFSE is not competitive with the proposed methods on any of the selected drug discovery datasets, as can be seen from Fig. 7B. As a sanity check for the algorithm, we include an instance where MFSE is trained only on high-fidelity data ('Only HF'), which performs similarly to our HF-only baseline. However, when used in the intended way ('HF + LF'), the performance does not consistently increase and even decreases in a number of instances. This inability to model both fidelities at the same time highlights the unique challenge posed by HTS drug discovery and the need for more effective transfer learning methods. At the same time, the ineffectiveness of MFSE when using both low- and high-fidelity labels shows that it is not trivial to incorporate low-fidelity information into a model even when multi-fidelity data is available.

## Extending to multiple fidelities

Our empirical analysis so far has assumed that transfer learning occurs between two clearly defined fidelity levels − 'low' to 'high'. This is traditionally the case for the majority of drug discovery by HTS projects and QMugs, the first large-scale multi-fidelity QM dataset, with the majority of the latter being single-fidelity. However, it is interesting to consider the case where more than two fidelities are present. Although this is not a common setting, we have selected the well-known QM7b dataset[47–49] which possesses HOMO and LUMO energy calculations at three different levels of theory: ZINDO, PBE0, and GW. For QM7b, as for the majority of datasets with more than two levels of fidelity[7] there is an 'ordering' of fidelities according to their precision: ZINDO < PBE0 < GW.

The strategies involving the inclusion of the low-fidelity labels or embeddings can be trivially extended to a setting with more than two fidelities. Concretely, a separate model is trained for each lower fidelity (here, ZINDO and PBE0), and the corresponding labels or embeddings are added to the high-fidelity model (here, GW) as usual. We have evaluated the direct inclusion of the low-fidelity labels, as well as embeddings generated by sum and adaptive readouts, both individually (i.e., only ZINDO or PBE0 at a time) and jointly (i.e., both labels or embeddings are concatenated to the internal molecular representation) and we report the results in Fig. 7A. Interestingly, the direct inclusion of the labels provides only small (less than 10% MAE decrease) or no performance uplifts, an observation shared with the outputs of the sum readout-based low-fidelity models. On the other hand, models leveraging neural embeddings provide significant improvements for ZINDO and PBE0 individually, despite the relatively low correlation between ZINDO and GW in particular (Fig. 7A). Furthermore, models jointly using ZINDO and PBE0 embeddings perform slightly better than PBE0. This is an interesting result as ZINDO is a relatively crude approach that is by definition more approximate than PBE0.

## Discussion

We have investigated the problem of learning an effective model for molecular property prediction on small-sample datasets typically encountered in the final stages of screening cascades characteristic of molecular design and drug discovery. Our focus was on showing the utility of transfer learning with graph neural networks for the multi-fidelity nature of this generative data process. More specifically, we have mainly focused on knowledge transfer between large-scale low-fidelity measurements that are inexpensive to obtain and sparse high-fidelity observations that are labour and resource-intensive. Typically, the effectiveness of the whole discovery process hinges on the assumption that one will be able to successfully select candidates for the final high-fidelity screening step. While prior work in materials chemistry has studied aspects of learning in multi-fidelity settings[7,38,41], here we tackle understudied and heterogeneous drug discovery by HTS tasks and quantum mechanics simulations through transfer learning with GNNs. We demonstrated a high level of generality through the successful application to quantum chemistry where we outperform existing methods, through explicitly evaluating transductive and inductive cases, through consistently strong results not only for our supervised VGAE architecture but also for 3D-aware networks like SchNet, and through extensions to more than two fidelities.

Our main algorithmic contribution lies in identifying and addressing the shortcomings of classical graph neural networks that are unable to harness the multi-fidelity observations produced by screening funnels. More specifically, we have proposed two main transfer learning schemes that enable effective knowledge transfer between fidelities: transfer by embeddings or predictions generated by models trained on low-fidelity data, and (supervised) pre-training on low-fidelity and fine-tuning on high-fidelity. All approaches are independent of the convolution operator used for feature extraction within graph neural networks. While learning molecular properties with architectures involving standard readouts is competitive on a range of tasks, particularly quantum properties that are extensive and not localised, this is generally not the case with large-scale and noisy drug discovery datasets. For those tasks, graph neural networks with adaptive readouts excel and unlock the transfer learning capability of supervised pre-training and fine-tuning, which is notoriously challenging for GNNs in general and molecular data in particular[50].

Our empirical analysis is extensive and covers several real-world datasets, various different baselines, and different problem domains and settings. The overall effectiveness and generality of the proposed approaches augur well for future applications of graph neural networks for transfer learning. More specifically, we envision impactful drug discovery applications in live high-throughput screening projects. The results in the inductive setting indicate that the massive amount of data collected during these campaigns can be transferred to predict high-fidelity activity for new compounds that did not exist during the HTS screening campaign, without needing to synthesise them first. Furthermore, from a resource and cost utilisation perspective, such projects operate on tight schedules and a fixed budget of high-fidelity evaluations (e.g., 10,000). Here, we have shown that transfer learning is particularly useful when a low amount of high-fidelity data is available (Fig. 6). Transfer learning can inform and improve the effectiveness of these costly steps, resulting in much more diverse and promising active molecules as well as lower costs to the discovery processes. In hybrid experimental and in silico workflows, as few as 500–1000 high-fidelity evaluations could be performed by traditional selection, with the rest of the budget invested into recommendations made with the help of transfer learning (operating in the inductive setting).

Ultimately, we hypothesise that multi-fidelity data and architectures are the natural step forward for a wide variety of molecular tasks specified by small-sample datasets. Transfer learning by embeddings is particularly interesting due to its high effectiveness and wide applicability. For instance, we have shown that the embeddings can successfully be used by models such as random forests and support vector machines, and envision applications to probabilistic methods such as Gaussian processes that can provide uncertainty estimates. Furthermore, the supervised variational graph autoencoder architecture has the potential to be useful in a generative setting, allowing a more informed and varied compound generation protocol since primary screens are designed to be diverse and are between 4 and 8 times larger than the commonly used ZINC dataset[51]. Another promising direction[32,52,53] might be to train low-fidelity predictors on multiple HTS projects at the same time, including potentially hundreds of protein targets, and aiming towards a 'universal' latent space organised by function or protein-ligand interactions. Extensions to other drug discovery technologies such as DNA-encoded molecule libraries are also an exciting direction.

## Methods

We start with a brief review of transfer learning and a formal description of our problem setting. This is followed by a section covering the preliminaries of graph neural networks (GNNs), including standard and adaptive readouts, as well as our supervised variational graph autoencoder architecture. Next, we formally introduce the considered transfer learning strategies, while also providing a brief overview of the frequently used approach for transfer learning in deep learning – a two stage learning mechanism consisting of pre-training and fine-tuning of a part or the whole (typically non-geometric) neural network[14]. In "Results" section, we perform an empirical study validating the effectiveness of the proposed approaches relative to the latter and state-of-the-art baselines for learning with multi-fidelity data.

## Overview of transfer learning and problem setting

Let $\mathcal{X}$ be an instance space and $X = \{x_1, \ldots, x_n\} \subset \mathcal{X}$ a sample from some marginal distribution $\rho_{\mathcal{X}}$. A tuple $\mathcal{D} = (\mathcal{X}, \rho_{\mathcal{X}})$ is called a domain. Given a specific domain $\mathcal{D}$, a task $\mathcal{T}$ consists of a label space $\mathcal{Y}$ and an objective predictive function $f : \mathcal{X} \to \mathcal{Y}$ that is unknown and needs to be learnt from training data given by examples $(x_i, y_i) \in \mathcal{X} \times \mathcal{Y}$ with $i = 1, \ldots, n$. To simplify the presentation, we restrict ourselves to the setting where there is a single source domain $\mathcal{D}_S$, and a single target domain $\mathcal{D}_T$. We also assume that $\mathcal{X}_T \subseteq \mathcal{X}_S$, and denote with $\mathcal{D}_S = \{(x_{S_1}, y_{S_1}), \ldots, (x_{S_n}, y_{S_n})\}$ and $\mathcal{D}_T = \{(x_{T_1}, y_{T_1}), \ldots, (x_{T_m}, y_{T_m})\}$, the observed examples from source and target domains. While the source domain task is associated with low-fidelity data, the target domain task is considered to be sparse and high-fidelity, i.e., it holds that $m \ll n$.

**Definition 1.** ([54,55]). Given a source domain $\mathcal{D}_S$ and a learning task $\mathcal{T}_S$, a target domain $\mathcal{D}_T$ and learning task $\mathcal{T}_T$, transfer learning aims to help improve the learning of the target predictive function $f_T$ in $\mathcal{D}_T$ using the knowledge in $\mathcal{D}_S$ and $\mathcal{T}_S$, where $\mathcal{D}_S \neq \mathcal{D}_T$ or $\mathcal{T}_S \neq \mathcal{T}_T$.

The goal in our problem setting is, thus, to learn the objective function $f_T$ in the target domain $\mathcal{D}_T$ by leveraging the knowledge from low-fidelity domain $\mathcal{D}_S$. The main focus is on devising a transfer learning approach for graph neural networks based on feature representation transfer. We propose extensions for two different learning settings: transductive and inductive learning. In the transductive transfer learning setup considered here, the target domain is constrained to the set of instances observed in the source dataset, i.e., $\mathcal{X}_T \subseteq \mathcal{X}_S$. Thus, the task in the target domain requires us to make predictions only at points observed in the source task/domain. In the inductive setting, we assume that source and target domains could differ in the marginal distribution of instances, i.e., $\rho_{\mathcal{X}_S} \neq \rho_{\mathcal{X}_T}$. For both learning settings, we assume that the source domain dataset is significantly larger as it is associated with low-fidelity simulations/approximations.

## Graph neural networks

Here, we follow the brief description of GNNs from[8]. A graph $G$ is represented by a tuple $G = (\mathcal{V}, \mathcal{E})$, where $\mathcal{V}$ is the set of nodes (or vertices) and $\mathcal{E} \subseteq \mathcal{V} \times \mathcal{V}$ is the set of edges. Here, we assume that the nodes are associated with feature vectors $\mathbf{x}_u$ of dimension $d$ for all $u \in \mathcal{V}$. The graph structure is represented by $\mathbf{A}$, the adjacency matrix of a graph $G$ such that $\mathbf{A}_{uv} = 1$ if $(u,v) \in \mathcal{E}$ and $\mathbf{A}_{uv} = 0$ otherwise. For a node $u \in \mathcal{V}$ the set of neighbouring nodes is denoted by $\mathcal{N}_u = \{v | (u, v) \in \mathcal{E} \vee (v, u) \in \mathcal{E}\}$. Assume also that a collection of graphs with corresponding labels $\{(G_i, y_i)\}_{i=1}^n$ has been sampled independently from a target probability measure defined over $\mathcal{G} \times \mathcal{Y}$, where $\mathcal{G}$ is the space of graphs and $\mathcal{Y} \subset \mathbb{R}$ is the set of labels. From now on, we consider that a graph $G$ is represented by a tuple $(\mathbf{X}_G, \mathbf{A}_G)$, with $\mathbf{X}_G$ denoting the matrix with node features as rows and $\mathbf{A}_G$ the adjacency matrix. The inputs of graph neural networks consist of such tuples, outputting predictions over the label space. In general, GNNs learn permutation invariant hypotheses that have consistent predictions for the same graph when presented with permuted nodes. This property is achieved through neighbourhood aggregation schemes and readouts that give rise to permutation invariant hypotheses. Formally, a function $f$ defined over a graph $G$ is called permutation invariant if there exists a permutation matrix $\mathbf{P}$ such that $f(\mathbf{P}\mathbf{X}_G, \mathbf{P}\mathbf{A}_G\mathbf{P}^\top) = f(\mathbf{X}_G, \mathbf{A}_G)$. The node features $\mathbf{X}_G$ and the graph structure (adjacency matrix) $\mathbf{A}_G$ are used to first learn representations of nodes $\mathbf{h}_v$, for all $v \in \mathcal{V}$. Permutation invariance in the neighbourhood aggregation schemes is enforced by employing standard pooling functions − sum, mean, or maximum. As succinctly described in[56], typical neighbourhood aggregation schemes characteristic of GNNs can be described by two steps:

$$\mathbf{a}_v^{(k)} = \text{AGGREGATE}(\{\mathbf{h}_u^{(k-1)} \mid u \in \mathcal{N}_v\}) \quad \text{and}$$
$$\mathbf{h}_v^{(k)} = \text{COMBINE}(\mathbf{h}_v^{(k-1)}, \mathbf{a}_v^{(k-1)}) \tag{1}$$

where $\mathbf{h}_u^{(k)}$ is a representation of node $u \in \mathcal{V}$ at the output of the $k^{\text{th}}$ iteration.

After $k$ iterations the representation of a node captures the information contained in its $k$-hop neighbourhood. For graph-level tasks such as molecular prediction, the last iteration is followed by a readout (also called pooling) function that aggregates the node features $\mathbf{h}_v$ into a graph representation $\mathbf{h}_G$. To enforce a permutation invariant hypotheses, it is again common to employ the standard pooling functions as readouts, namely sum, mean, or maximum.

## Graph neural networks with adaptive (neural) readouts

Standard readout functions (i.e., sum, mean, and maximum) in graph neural networks do not have any parameters and are, thus, not amenable for transfer learning between domains. Motivated by this, we build on our recent work[8] that proposes a neural network architecture to aggregate learnt node representations into graph embeddings. This allows for freezing the part of a GNN architecture responsible for learning effective node representations and fine-tuning the readout layer in small-sample downstream tasks. In the remainder of the section, we present a Set Transformer readout that retains the permutation invariance property characteristic of standard pooling functions. Henceforth, suppose that after completing a pre-specified number of neighbourhood aggregation iterations, the resulting node features are collected into a matrix $\mathbf{H} \in \mathbb{R}^{M \times D}$, where $M$ is the maximal number of nodes that a graph can have in the dataset and $D$ is the dimension of the output node embedding. For graphs with less than $M$ vertices, $\mathbf{H}$ is padded with zeros.

Recently, an attention-based neural architecture for learning on sets has been proposed by Lee et al.[57]. The main difference compared to the classical attention model proposed by Vaswani et al.[9] is the absence of positional encodings and dropout layers. As graphs can be seen as sets of nodes, we leverage this architecture as a readout function in graph neural networks. For the sake of brevity, we omit the details of classical attention models[9] and summarise only the adaptation to sets (and thus graphs). The Set Transformer (ST) takes as input matrices with set items (in our case, graph nodes) as rows and generates graph representations by composing encoder and decoder modules implemented using attention:

$$\text{ST}(\mathbf{H}) = \frac{1}{K} \sum_{k=1}^{K} [\text{Decoder}(\text{Encoder}(\mathbf{H}))]_k \tag{2}$$

where $[\cdot]_k$ refers to a computation specific to head $k$ of a multi-head attention module. The encoder-decoder modules follow the definition of Lee et al.[57]:

$$\text{Encoder}(\mathbf{H}) = \text{MAB}^n(\mathbf{H}, \mathbf{H}) \tag{3}$$

$$\text{Decoder}(\mathbf{Z}) = \text{FF}(\text{MAB}^m(\text{PMA}(\mathbf{Z}), \text{PMA}(\mathbf{Z}))) \tag{4}$$

$$\text{PMA}(\mathbf{Z}) = \text{MAB}(\mathbf{s}, \text{FF}(\mathbf{Z})) \tag{5}$$

$$\text{MAB}(\mathbf{X}, \mathbf{Y}) = \mathbf{A} + \text{FF}(\mathbf{A}) \tag{6}$$

$$\mathbf{A} = \mathbf{X} + \text{MultiHead}(\mathbf{X}, \mathbf{Y}, \mathbf{Y}). \tag{7}$$

Here, $\mathbf{H}$ denotes the node features after neighbourhood aggregation and $\mathbf{Z}$ is the encoder output. The encoder is a chain of $n$ classical multi-head attention blocks (MAB) without positional encodings. The decoder component consists of a pooling by multi-head attention block (PMA) (which uses a learnable seed vector $\mathbf{s}$ within a multi-head attention block to create an initial readout vector) that is further processed via a chain of $m$ self-attention modules and a linear projection block (also called feedforward, FF). In contrast to typical set-based neural architectures that process individual items in isolation (most notably deep sets[58]), the presented adaptive readouts account for interactions between all the node representations

generated by the neighbourhood aggregation scheme. A particularity of this architecture is that the dimension of the graph representation can be disentangled from the node output dimension and the aggregation scheme.

## Supervised variational graph autoencoders

We start with a review of variational graph autoencoders (VGAEs), originally proposed by Kipf and Welling[59], and then introduce a variation that allows for learning of a predictive model operating in the latent space of the encoder. More specifically, we propose to jointly train the autoencoder together with a small predictive model (multi-layer perceptron) operating in its latent space by including an additional loss term that accounts for the target labels. Below, we follow the brief description of [6].

A variational graph autoencoder consists of a probabilistic encoder and decoder, with several important differences compared to standard architectures operating on vector-valued inputs. The encoder component is obtained by stacking graph convolutional layers to learn the parameter matrices $\boldsymbol{\mu}$ and $\boldsymbol{\sigma}$ that specify the Gaussian distribution of a latent space encoding. More formally, we have that

$$q(\mathbf{Z} \mid \mathbf{X}, \mathbf{A}) = \prod_{i=1}^{N} q(\mathbf{z}_i \mid \mathbf{X}, \mathbf{A}) \quad \text{and} \quad q(\mathbf{z}_i \mid \mathbf{X}, \mathbf{A}) = \mathcal{N}(\mathbf{z}_i \mid \boldsymbol{\mu}_i, \operatorname{diag}(\boldsymbol{\sigma}_i^2)),$$
(8)

with $\boldsymbol{\mu} = \mathrm{GCN}_{\mu,n}(\mathbf{X}, \mathbf{A})$ and $\log \boldsymbol{\sigma} = \mathrm{GCN}_{\sigma,n}(\mathbf{X}, \mathbf{A})$. Here, $\mathrm{GCN}_{\cdot,n}$ is a graph convolutional neural network with $n$ layers, $\mathbf{X}$ is a node feature matrix, $\mathbf{A}$ is the adjacency matrix of the graph, and $\mathcal{N}$ denotes the Gaussian distribution. Moreover, the model typically assumes the existence of self-loops, i.e., the diagonal of the adjacency matrix consists of ones.

The decoder reconstructs the entries in the adjacency matrix by passing the inner product between latent variables through the logistic sigmoid. More formally, we have that

$$p(\mathbf{A} \mid \mathbf{Z}) = \prod_{i=1}^{N} \prod_{j=1}^{N} p(\mathbf{A}_{ij} \mid \mathbf{z}_i, \mathbf{z}_j) \quad \text{and} \quad p(\mathbf{A}_{ij} = 1 \mid \mathbf{z}_i, \mathbf{z}_j) = \tau(\mathbf{z}_i^{\top} \mathbf{z}_j),$$
(9)

where $\mathbf{A}_{ij}$ are entries in the adjacency matrix $\mathbf{A}$ and $\tau(\cdot)$ is the logistic sigmoid function. A variational graph autoencoder is trained by optimising the evidence lower-bound loss function that can be seen as the combination of a reconstruction and a regularisation term:

$$\tilde{\mathcal{L}}(\mathbf{X}, \mathbf{A}) = \underbrace{\mathbb{E}_{q(\mathbf{Z} \mid \mathbf{X}, \mathbf{A})}\left[\log p(\mathbf{A} \mid \mathbf{Z})\right]}_{\mathcal{L}_{\text{RECON}}} - \underbrace{\mathrm{KL}\left[q(\mathbf{Z} \mid \mathbf{X}, \mathbf{A}) \parallel p(\mathbf{Z})\right]}_{\mathcal{L}_{\text{REG}}}$$
(10)

where $\mathrm{KL}[q(\cdot) \| p(\cdot)]$ is the Kullback-Leibler divergence between the variational distribution $q(\cdot)$ and the prior $p(\cdot)$. The prior is assumed to be a Gaussian distribution given by $p(\mathbf{Z}) = \prod_i p(\mathbf{z}_i) = \prod_i \mathcal{N}(\mathbf{z}_i \mid 0, \mathbf{I})$. As the adjacency matrices of graphs are typically sparse, instead of taking all the negative entries when training one typically performs sub-sampling of entries with $\mathbf{A}_{ij} = 0$.

We extend this neural architecture by adding a feedforward component operating on the latent space and account for its effectiveness via the mean squared error loss term that is added to the optimisation objective. More specifically, we optimise the following loss function:

$$\mathcal{L}(\mathbf{X}, \mathbf{A}, \mathbf{y}) = \tilde{\mathcal{L}}(\mathbf{X}, \mathbf{A}) + \frac{1}{N} \sum_{i=1}^{N} \| \nu(\mathbf{Z}_i) - \mathbf{y}_i \|^2,$$
(11)

where $\nu(\mathbf{Z})$ is the predictive model operating on the latent space embedding $\mathbf{Z}$ associated with graph $(\mathbf{X}, \mathbf{A})$, $\mathbf{y}$ is the vector with target labels, and $N$ is the number of labelled instances. Figure 2 illustrates the

setting and our approach to transfer learning using supervised variational graph autoencoders.

We note that our supervised variational graph autoencoder resembles the joint property prediction variational autoencoder (JPP-VAE) proposed by Gómez-Bombarelli et al.[39]. Their approach has been devised for generative purposes, which we do not consider here. The main difference to our approach, however, is the fact that JPP-VAE is a sequence model trained directly on the SMILES[60] string representation of molecules using recurrent neural networks, a common approach in generative models[61,62]. The transition from traditional VAEs to geometric deep learning (graph data) in the first place, and then to molecular structures is not a trivial process for at least two reasons. Firstly, a variational graph autoencoder only reconstructs the graph connectivity information (i.e., the equivalent of the adjacency matrix) and not the node (atom) features, according to the original definition by Kipf and Welling. This is in contrast to traditional VAEs where the latent representation is directly optimised against the actual input data. The balance between reconstruction functions (for the connectivity, and node features respectively) is thus an open question in geometric deep learning. Secondly, for molecule-level tasks such as prediction and latent space representation, the readout function of the variational graph autoencoders is crucial. As we have previously explored in[8] and further validate in "Results" section, standard readout functions such as sum, mean, or maximum lead to uninformative representations that are similar to completely unsupervised training (i.e., not performing well in transfer learning tasks). Thus, the supervised or guided variational graph autoencoders presented here are also an advancement in terms of graph representation learning for modelling challenging molecular tasks at the multi-million scale.

## Feature augmentation via low-fidelity simulations

In the context of quantum chemistry and the design of molecular materials, the most computationally demanding task corresponds to the calculation of energy contribution that constitutes only a minor fraction of total energy, while the majority of the remaining calculations can be accounted for via efficient proxies[28]. Motivated by this, Ramakrishnan et al.[28] have proposed an approach known as Δ-machine learning, where the desired molecular property is approximated by learning an additive correction term for a low-fidelity proxy. For linear models, an approach along these lines can be seen as feature augmentation where instead of the constant bias term one appends the low-fidelity approximation as a component to the original representation of an instance. More specifically, if we represent a molecule in the low-fidelity domain via $\mathbf{x} \in \mathcal{X}_S$ then the representation transfer for $\mathcal{D}_T$ can be achieved via the feature mapping

$$\Psi_{\text{Label}}(\mathbf{x}) = \| (f_S(\mathbf{x}), \mathbf{x})$$
(12)

where $\|(\cdot, \cdot)$ denotes concatenation in the last tensor dimension and $f_S$ is the objective prediction function associated with the source (low-fidelity) domain $\mathcal{D}_S$ defined in "Overview of transfer learning and problem setting" section. We consider this approach in the context of transfer learning for general methods (including GNNs) and standard baselines that operate on molecular fingerprints (e.g., support vector machines, random forests, etc.). A limitation of this approach is that it constrains the high-fidelity domain to the transductive setting and instances that have been observed in the low-fidelity domain. A related set of methods in the drug discovery literature called high-throughput fingerprints[34–37] function in effectively the same manner, using a vector of hundreds of experimental single-dose (low-fidelity) measurements and optionally a standard molecular fingerprint as a general molecular representation (i.e., not formulated specifically for transductive or

multi-fidelity tasks). In these cases, the burden of collecting the low-fidelity representation is substantial, involving potentially hundreds of experiments (assays) that are often disjoint, resulting in sparse fingerprints and no practical way to make predictions about compounds that have not been part of the original assays. In drug discovery in particular it is desirable to extend beyond this setting and enable predictions for arbitrary molecules, i.e., outside of the low-fidelity domain. Such a model would enable property prediction for compounds before they are physically synthesised, a paradigm shift compared to existing HTS approaches. To overcome the transductive limitation, we consider a feature augmentation approach that leverages low-fidelity data to learn an approximation of the objective function in that domain. Then, transfer learning to the high-fidelity domain happens via the augmented feature map

$$\Psi_{(\text{Hybrid label})}(\mathbf{x}) = \begin{cases} \| (f_S(\mathbf{x}), \mathbf{x}) & \text{if} \quad \mathbf{x} \in \mathcal{X}_S, \\ \| (\tilde{f}_S(\mathbf{x}), \mathbf{x}) & \text{otherwise} \end{cases} \quad (13)$$

where $\tilde{f}_S$ is an approximation of the low-fidelity objective function $f_S$. This is a hybrid approach that allows extending to the inductive setting with a different treatment between instances observed in the low-fidelity domain and the ones associated with the high-fidelity task exclusively. Another possible extension that treats all instances in the high-fidelity domain equally is via the map $\Psi_{(\text{Predicted label})}$ that augments the input feature representation using an approximate low-fidelity objective ($\tilde{f}_S$), i.e.,

$$\Psi_{(\text{Predicted label})}(\mathbf{x}) = \| (\tilde{f}_S(\mathbf{x}), \mathbf{x}) \quad (14)$$

Our final feature augmentation amounts to learning a latent representation of molecules in the low-fidelity domain using a supervised autoencoder (see "Supervised variational graph autoencoders" section), then jointly training alongside the latent representation of a model that is being fitted to the high-fidelity data. This approach also lends itself to the inductive setting. More formally, transfer learning in this case can be achieved via the feature mapping

$$\Psi_{\text{Embeddings}}(\mathbf{x}) = \| (\psi_S(\mathbf{x}), \psi_T(\mathbf{x})) \quad (15)$$

where $\psi_S(\mathbf{x})$ is the latent embedding obtained by training a supervised autoencoder on low-fidelity data $\mathcal{D}_S$, and $\psi_T(\mathbf{x})$ represents the latent representation of a model trained on the sparse high-fidelity task. Note that $\psi_S(\mathbf{x})$ is fixed (the output of the low-fidelity model which is trained separately), while $\psi_T(\mathbf{x})$ is the current embedding of the high-fidelity model that is being learnt alongside $\psi_S(\mathbf{x})$ and can be updated.

### Pre-training and fine-tuning of graph neural networks
Supervised pre-training and fine-tuning is a transfer learning strategy that has previously proven successful for non-graph neural networks in the context of energy prediction for small organic molecules. In its simplest form, and as previously used by Smith et al.[14], the strategy consists of first training a model on the low-fidelity data $\mathcal{D}_S$ (the pre-training step). Afterwards, the model is retrained on the high-fidelity data $\mathcal{D}_T$, such that it now outputs predictions at the desired fidelity level (the fine-tuning step). For the fine-tuning step, certain layers of the neural network are typically frozen, which means that gradient computation is disabled for them. In other words, their weights are fixed to the values learnt during the pre-training step and are not updated. This technique reduces the number of learnable parameters, thus helping to avoid over-fitting to a smaller high-fidelity dataset and reducing training times. Formally, we assume that we have a low-fidelity predictor $\tilde{f}_S$ (corresponding to pre-training) and define the steps required to re-train or fine-tune a blank model $\tilde{f}_{T_0}$ (in domain $\mathcal{T}$)

into a high-fidelity predictor $\tilde{f}_T$

$$\mathbf{W}_S = \text{Weights}(\tilde{f}_S) \quad (\text{Extract weights of pre-trained model } \tilde{f}_S) \quad (16)$$

$$\mathbf{W}_S = \text{Freeze}(\mathbf{W}_{S_{\text{GCN}}}, \ldots) \quad (\text{Freeze components, e.g. GCN layers}) \quad (17)$$

$$\tilde{f}_{T_0} = \mathbf{W}_S \quad (\text{Assign weights of } \tilde{f}_S \text{ to a blank model } \tilde{f}_{T_0}) \quad (18)$$

where $\tilde{f}_{T_0}$ is fine-tuned into $\tilde{f}_T$. As a baseline, we define a simple equivalent to the neural network in Smith et al., where we pre-train and fine-tune a supervised VGAE model with the sum readout and without any frozen layers. This is justified by GNNs having a small number of layers to avoid well-known problems such as oversmoothing. As such, the entire VGAE is fine-tuned and the strategy is termed $\Psi_{(\text{Tune VGAE})}$:

$$\mathbf{W}_S = \text{Freeze}(\varnothing) \quad (\text{No component is frozen}) \quad (19)$$

$$\tilde{f}_{T_0} = \mathbf{W}_S \quad (\text{Assign initial weights}) \quad (20)$$

$$\Psi_{(\text{Tune VGAE})}(\mathbf{x}) = \tilde{f}_T(\mathbf{x}) \quad (\text{The final model is the fine-tuned } \tilde{f}_T) \quad (21)$$

Standard GNN readouts such as the sum operator are fixed functions with no learnable parameters. In contrast, adaptive readouts are implemented as neural networks, and the overall GNN becomes a modular architecture composed of (1) the supervised VGAE layers and (2) an adaptive readout. Consequently, there are three possible ways to freeze components at this level: (i) frozen graph convolutional layers and trainable readout, (ii) trainable graph layers and frozen readout, and (iii) trainable graph layers and trainable readout (no freezing). After a preliminary study on a representative collection of datasets, we decided to follow strategy (i) due to empirically strong results and overall originality for transfer learning with graph neural networks. More formally, we have that

$$\mathbf{W}_S = \text{Freeze}(\mathbf{W}_{S_{\text{GCN}}}) \quad (\text{Freeze all GCN layers}) \quad (22)$$

$$\tilde{f}_{T_0} = \mathbf{W}_S \quad (\text{Assign initial weights}) \quad (23)$$

$$\Psi_{(\text{Tune readout})}(\mathbf{x}) = \tilde{f}_T(\mathbf{x}) \quad (\text{The final model is the fine-tuned } \tilde{f}_T) \quad (24)$$

### Data selection and filtering
For drug discovery tasks, low-fidelity (LF) data consists of single-dose measurements (SD, performed at a single concentration) for a large collection of compounds. The high-fidelity (HF) data consists of dose-response (DR) measurements corresponding to multiple different concentrations that are available for a small collection of compounds (see Fig. 1, top). In the quantum mechanics experiments, we have opted for the recently-released QMugs dataset with 657K unique drug-like molecules and 12 quantum properties. The data originating from semi-empirical GFN2-xTB simulations act as the low-fidelity task, and the high-fidelity component is obtained via density-functional theory (DFT) calculations ($\omega$B97X-D/def2-SVP). The resulting multi-fidelity datasets are defined as datasets where SMILES-encoded molecules are associated with two different measurements of different fidelity levels.

As modelling large-scale high-throughput screening data and transfer learning in this context are understudied applications, a significant effort was made to carefully select and filter suitable data from public (PubChem) and proprietary (AstraZeneca) sources, covering a multitude of different settings. To this end, we have assembled several multi-fidelity drug discovery datasets (Fig. 1, top) from PubChem, aiming to capture the heterogeneity intrinsic to large-scale screening campaigns, particularly in terms of assay types, screening technologies, concentrations, scoring metrics, protein targets, and scope. This

has resulted in 23 multi-fidelity datasets (Supplementary Table 1) that are now part of the concurrently published MF-PCBA collection[29]. We have also curated 16 multi-fidelity datasets based on historical Astra-Zeneca (AZ) HTS data (Supplementary Table 2), the emphasis now being put on expanding the number of compounds in the primary (1 million+) and confirmatory screens (1000 to 10,000). The search, selection, and filtering steps, along with the naming convention are detailed in Supplementary Notes 5 and[29]. As the QMugs dataset contains a few erroneous calculations, we apply a filtering protocol similar to the drug discovery data and remove the values that diverge by more than 5 standard deviations, which removes just over 1% of the molecules present. The QMugs properties are listed in Supplementary Table 3. For the transductive setting, we selected a diverse and challenging set of 10K QMugs molecules (Supplementary Notes 5.1), which resembles the drug discovery setting.

While methods to artificially generate multi-fidelity data with desired fidelity correlations have recently been proposed[63], we did not pursue this direction as remarkably large collections of real-world multi-fidelity data are available, covering a large range of fidelity correlations and diverse chemical spaces. Furthermore, the successful application of such techniques to molecular data is yet to be demonstrated.

### Reporting summary
Further information on research design is available in the Nature Portfolio Reporting Summary linked to this article.

## Data availability
The MF-PCBA, QMugs, and QM7b datasets are publicly available and accessible by following the instructions presented in their respective papers. We provide additional instructions relevant to our workflow in our code repository. The proprietary AstraZeneca HTS data collection is not publicly available. For the purposes of this work, the proprietary data are pre-processed and used within our computational workflow following identical steps to MF-PCBA. Source data are provided with this paper (for all Figures and Supplementary Figs., with the exception of the UMAP plots of Fig. 3C). Due to the large size, the UMAP data is instead hosted on the GitHub repository listed below.

## Code availability
The source code that enables all experiments to be reproduced and the instructions for accessing the datasets are hosted on GitHub[64]: https://github.com/davidbuterez/multi-fidelity-gnns-for-drug-discovery-and-quantum-mechanics.

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

## Acknowledgements

We are thankful for the PhD sponsorship awarded to D.B. and access to the Scientific Computing Platform within AstraZeneca.

## Author contributions

D.B., J.P.J., S.J.K, D.O., and P.L. were involved in conceptualising and reviewing the manuscript. D.B. designed and carried out experiments and produced the figures and tables. J.P.J, S.J.K, D.O., and P.L. jointly supervised the work.

## Competing interests

D.B. has been supported by a fully funded PhD sponsorship from AstraZeneca. J.P.J, S.J.K, and D.O. are employees and shareholders at AstraZeneca. The remaining authors declare no competing interests.
