## [Peer Review File · Nature Communications]

Transfer learning with graph neural networks for improved molecular property prediction in the multi-fidelity settingReviewers' comments:

Reviewer #1 (Remarks to the Author):

Buterez et al. describe a systematic study of multi-fidelity transfer-learning approaches in the context of drug/molecular discovery problems. A wide range of transfer learning techniques are investigated across a number of datasets. In the big data era with more and more datasets of mixed fidelities appearing, as well as expressive and general machine learning models for molecules such as GNNs, transfer learning or pre-training has become an important topic in the field; consequently this is a timely paper. However, there are some major concerns. One concern is the pre-training methods primarily help a model using a specific readout function, but are significantly less effective for a more common model using sum pooling, which can lead one to question if these approaches can really be helpful to other GNN models which do not have this specific readout. Another concern is that in many cases, the transfer learning approaches either fail to beat a simple direct approach of using low fidelity labels, or in fact harm the performance of the predictions, which represents a significant risk in using transfer learning, which is not well discussed or addressed. Overall, the paper may be publishable but should address these concerns and additional ones below.

-It seems for the drug dataset used, the GNN model used with sum pooling is more or less completely ineffective for the prediction of properties. From that perspective, demonstrating the effectiveness of transfer learning may get lost in the noise, so to speak. Also, much of the benefit of the transfer learning techniques highlighted really only occur with the custom adaptive pooling function, which is not widely used. As it stands, it is hard for an observer to evaluate if this work demonstrates the effectiveness of transfer learning, especially as most GNN models have some kind of sum or mean pooling.

-It also appears using the low-fidelity label directly beats most transfer learning methods handily, especially for the QM9 type dataset in supplementary Figure 7. More discussion is warranted here, and why the fancier and more complex approaches cannot outperform a simple and direct baseline.

-In more than a few cases, transfer learning hurts the performance of the models significantly. This is a very concerning phenomenon as one should expect that with the additional effort of procuring low fidelity labels and additional compute for training, that at the very least this should not harm the performance. This is perhaps not discussed as much as it should be, as it represents a major potential risk in transfer learning, and it does not seem any of the proposed transfer learning methods can reliably avoid this risk.

-There are concerns regarding the generalizability of this approach and the conclusions in this paper. The work is limited to a set of simple GNN architectures and a non-conventional pooling function. While the authors claim that different model architectures did not have a significant impact on the conclusions in the supplementary info., no convincing evidence is shown to support this argument. While understandably this involves significant additional compute for experiments, the current examples may not be entirely convincing to readers.

-Some discussion is perhaps warranted on unsupervised or self-supervised pre-training, which can provide performance benefits with no low-fidelity labels. Certainly, in some cases where the supervised pre-training benefits are minor or detrimental, one may wonder if the unsupervised approach may be more effective.

Other minor issues:

-Figure 1 and 2 could be more illustrative for more general audiences and currently difficult to follow. It would also help if there was a figure or subfigure summarizing the 9 different methods in Figure 4 somewhere.

-Dataset mean should be provided for the QMugs dataset in Figure 3 for consistency

-Figure labels should be consistent and more clearly mentioned in the text. I.e. in Figure 4, is tune readout in 4A the same as tune vgae in 4b? If so, they should use the same name. Section 3 and more specifically 3.2 should use or mention the Figure 4a label names, or else it is difficult to track which method is which.

-The % MAE decrease in Figure 4 is confusing - why not use % improvement like in Figure 5?

-Not clear what "no low-fidelity set" means in Figure 5

-The switching between high fidelity and low fidelity with single dose and dose response labels are confusing

Reviewer #2 (Remarks to the Author):

This is a well written paper, which addresses an emerging area in informatics, namely how can one better leverage all of the information which is present for a particular task, rather than being restricted to a single fidelity of data, selected either by volume or quality. This is important as these two selection criteria are often in opposition – i.e. where there is large data volume, it is often the case that the quality (or more accurately the fidelity, since the measurements are often made well, but using techniques with a lot of associated noise or assumptions) is reduced.

The authors treat the problem as a transfer learning task, transferring 'knowledge' (as it were) gained from a higher volume, lower fidelity, task to a model which includes information gleaned from higher fidelity data. It should be noted that as a parallel submission to this effort, the authors have made available a set of benchmark datasets, which are of great value to the community.

One theme that does persist through the paper, perhaps due to the authors' key areas of expertise, is somewhat of a lack of appreciation for the literature on multi-fidelity machine learning and optimization in the materials chemistry domain. Some examples which the authors might find helpful to consider contrasting and comparing their approach to could be:

1. Yang, C.-H. et al. Multi-fidelity machine learning models for structure–property mapping of organic electronics. *Computational Materials Science* 213, 111599 (2022).

2. Patra, A. et al. A multi-fidelity information-fusion approach to machine learn and predict polymer bandgap. *Computational Materials Science* 172, 109286 (2020).

3. Meng, X. & Karniadakis, G. E. A composite neural network that learns from multi-fidelity data: Application to function approximation and inverse PDE problems. *Journal of Computational Physics* 401, 109020 (2020).

4. Chen, C., Zuo, Y., Ye, W., Li, X. & Ong, S. P. Learning properties of ordered and disordered materials from multi-fidelity data. *Nat Comput Sci* 1, 46–53 (2021).

5. Fare, C., Fenner, P., Benatan, M., Varsi, A. & Pyzer-Knapp, E. O. A multi-fidelity machine learning approach to high throughput materials screening. *npj Computational Materials* 8, 1–9 (2022).

It is my opinion that publication of this type of work in a journal such as *Nature Communications* requires that there be first of a kind novelty beyond a specific domain (such as drug discovery), or significant performance improvements with real world consequences. Without the consideration of methods beyond those applied specifically to drug discovery, it is hard for me as a reviewer to make that judgement.

The authors tend to treat each low-fidelity data source as an augmentation to a single model. I am unsure as to how this approach would naturally deal with, for example, two different ways of measuring the same target. This is common in quantum chemistry, for example. Since quantum chemistry is one of the use cases here, I would suggest that the authors consider how to expand their work to include these kinds of task. This would imply that the authors be able to cover the following questions:

(a) the effectiveness of transfer learning when there are more than two levels (e.g. high and low). A comparison of a strategy where models are built through stepwise pairing and when models are built through many-to-one transfer would be helpful. This would provide strength to the main value proposition of the paper, namely that one should use all available data, rather than a single fidelity.

(b) The interplay between different fidelities or augmentations into the model. When more than two fidelities are investigated, there is much to be learned from the interactions between two non-target fidelities which can inform, for example, further data collection. An example may be to understand how electronic properties calculated at one combination of functional/approach and basis set might relate to another.

(c) The response of a model in the multi-fidelity setting, where one of the lower fidelities are non-correlated (i.e. poisonous to the high fidelity model), especially in areas of configuration space. An example of this is the use of simulated data, where the simulation model may have been tuned for a subset of chemical space and has degraded performance outside of this space.

It should be noted that principled methods for generating synthetic fidelities based on 'anchoring' fidelities exist, with one recent example being:

Fare, C., Fenner, P. & Pyzer-Knapp, E. O. A Principled Method for the Creation of Synthetic Multi-fidelity Data Sets. Preprint at <https://doi.org/10.48550/arXiv.2208.05667> (2022).

I found it unclear from the paper as to the requirement for complete data coverage over the low-fidelity tasks. That is, if there are multiple data sources with different 'fidelities' of data, does the training model need to have complete data sets to train, or can it deal with partial data across the multi-fidelity regime? I would suggest that being able to deal with multiple fidelities implies that the model should really be able to deal with datasets where there is not complete coverage of targets at all fidelities to maximise the impact of the method. It would be good for the authors to directly address this point.

In summary, I think this is an interesting paper which has some promise. Before I can recommend its acceptance and publication in a journal such as Nature Communications, I would require that the questions I have raised, namely around comparisons to existing literature from the materials chemistry domain, and the applicability of the proposed method to more than 'high/low' fidelity classification and the requirements for data completeness in the low fidelity regimes, be addressed.

Reviewer #3 (Remarks to the Author):

In current manuscript, authors described various methodologies for infusing models learned from dense low fidelity data with models learned from sparse high fidelity data to increase model performance on sparse high fidelity data. The authors tried various fusion strategies as well as different training settings (transductive vs inductive setting). The included datasets are quite extensive including both industry and open source. Although it is interesting to see how different fusion strategy can improve the performance comparing with model learned on sparse high fidelity data alone, I feel that both the applied fusion strategy and neural network architecture are not novel enough to be accepted for this journal.

A few comments for authors to consider to improve the quality of manuscript:

1) The authors are quite ambitious to make a systematic survey. But, in my opinion, considering too

many scenario/strategies makes the manuscript hard to follow. It is unclear to me in the end, which fusion strategy is the best one to recommend. The authors can probably consider to simplify the structure of the manuscript.

2) It seems some strategies were mentioned earlier, but it was not listed and discussed later (at least clearly matched to the methods in Figure 4). For example, "weight transfer as in Smith et al." was mentioned as a strategy for comparison. It is unclear to me which augmentation method in Figure 4, 5 is for this strategy (?).

3) The authors compared fused models with models without fusion. But model structure of non-fusion model is the same with those fused models. This is perhaps not surprising since more low fidelity data was implicitly included in the training on high fidelity model. What about comparing with non-fusion models using other network architecture?

We would like to first of all thank the Reviewers for their constructive and insightful feedback. Thanks to them, we are now able to submit a substantially revised manuscript that addresses the Reviewers' concerns and delves further than their initial inquiries. All of the major sections, including the title, abstract, introduction, methods, results, and discussion have been significantly reworked to account for the new experiments and suggestions of the Reviewers, in this process clarifying our theoretical and experimental contributions. Moreover, a new **Related work** section is dedicated to a better contextualisation of our work. We also summarise the sources of novelty in the following section, answering concerns of Reviewer 1 and 3.

Novelty statement

First of all, we believe that it is just as important to consider the algorithmic contributions which are foundational to graph representation learning. We empirically demonstrated the limited potential of standard GNN architectures as part of different transfer learning strategies and identified the bottleneck: the readout function. We have extensively discussed the interplay between the different readout functions in our response to Reviewer 1's comments (below). Most importantly, our proposed adaptive readouts unlock a powerful and novel transfer learning strategy for GNNs: supervised pre-training and fine-tuning of the readout function, which is similar in scope and usability to the prevalent pre-training schemes used in Large Language Models, but has eluded GNNs so far. Pre-training and fine-tuning GNNs is a notoriously difficult problem [Hu+20] that has been known for a few years, particularly for molecular tasks, which also explains the relatively scarce literature for this type of transfer learning for GNNs. We also introduce a supervised VGAE model that leverages our adaptive readout functions. While being constructed of previously-developed components, it is a novel model architecture for multi-fidelity learning that is observed to offer substantial benefits for transfer learning.

Our contribution is not limited in scope to a single domain. While we have focused on drug discovery, generality was one of the aims for our framework. We extensively cover multi-fidelity learning for the quantum mechanics dataset QMugs, the only multi-fidelity quantum dataset of comparable scale to our drug discovery data [Ise+22]. Our analysis spans a number of quantum properties of different types, which are tackled both *transductively* and *inductively*. Remarkably, although transfer learning is more established in the field of quantum machine learning, we are still able to outperform existing strategies when using the strategies developed here. Secondly, in our revised evaluation we extend our strategies to cover cases with more than 2 fidelities. We take this opportunity to also demonstrate that transfer learning as presented here also works for SchNet [Sch+17; Sch+18], and thus for the widely-used family of deep learning models operating at the level of 3D atomic coordinates and physically-relevant quantities like bond distances and angles. In summary, our proposed methodologies show benefit across a large number of molecular properties, settings and model types.

We show significant improvements that we believe have substantial potential impacts on the real world. Within drug discovery, our approach is a completely novel application and we utilise public as well as commercial HTS data from historical drug discovery campaigns. This data has not, to our knowledge, been investigated in a multi-fidelity setting, despite being a core technology in the pharmaceutical industry. As such, there are potential direct benefits for real-world drug discovery projects which were previously unexplored. We would also like to explicitly distinguish between the *transductive* (high- and low-fidelity data available for all points) and the more challenging *inductive* settings (with only partial low-fidelity coverage). The transductive setting acts as an excellent benchmark for molecular transfer learning, as having both fidelities enables the comparison with an 'ideal' baseline where the actual low-fidelity label is available, and it is the approach taken in previous multi-fidelity studies, including the references suggested by the Reviewer. However, the more challenging *inductive* setting has generally been neglected, despite being of direct practical relevance: when low-fidelity experimental labels are required for a method, it cannot be used to assess which molecules to make or purchase, since these costs and latency will outstrip any difference in the cost of different assays. This scenario is virtually always seen in drug discovery by HTS projects, since the primary (low-fidelity) screening stage is followed by an optimisation where new molecules are designed, made and tested.

In this context, we emphasise that our extensive evaluation reveals that no previous multi-fidelity or transfer learning techniques works for the drug discovery by HTS datasets. In contrast, we are able to report uplifts in predictive performance using our new methods between 20% and 80% depending on the dataset, and severalfold in the best cases, for both *transductive* and *inductive* settings. Our most striking result is that transfer learning as presented here can improve the accuracy of predictive models

by up to 8 times while using an order of magnitude less high-fidelity training data (**Results, Section 4.4** and **Figure 6A**). In the case of drug discovery, transfer learning by embeddings learnt in an inductive setting led to the same performance as the low-fidelity labels themselves (usable only in the transductive setting). This is a remarkable result as the low-fidelity information must be extrapolated from models that have not encountered the molecules before, indicating strong generalisation performance.

While we are not able to disclose impacts of these methods on active commercial projects, this would constitute an unreasonable standard for impact that very few academic papers would meet. Our ultimate vision for the impact of this work in the real-world is to integrate transfer learning into wet-lab experimental protocols, allowing new, promising compounds to be investigated (which would otherwise be missed by traditional methods), cost savings by a more focused and less wasteful experimental validation, and allowing new capabilities that transcend single projects, for example by training a ‘universal’ *in silico* low-fidelity simulator trained on hundreds of project assays, or adapting our variational architectures to generative uses. Several of these ideas are being trialled internally and we believe that we have made the first big step forward through the work presented here.

Reviewer 1

1. *Buterez et al. describe a systematic study of multi-fidelity transfer-learning approaches in the context of drug/molecular discovery problems. A wide range of transfer learning techniques are investigated across a number of datasets. In the big data era with more and more datasets of mixed fidelities appearing, as well as expressive and general machine learning models for molecules such as GNNs, transfer learning or pre-training has become an important topic in the field; consequently this is a timely paper. However, there are some major concerns. One concern is the pre-training methods primarily help a model using a specific readout function, but are significantly less effective for a more common model using sum pooling, which can lead one to question if these approaches can really be helpful to other GNN models which do not have this specific readout. Another concern is that in many cases, the transfer learning approaches either fail to beat a simple direct approach of using low fidelity labels, or in fact harm the performance of the predictions, which represents a significant risk in using transfer learning, which is not well discussed or addressed. Overall, the paper may be publishable but should address these concerns and additional ones below.*

Response

Firstly, we would like to highlight the positive attributes of our work that the Reviewer has identified in this first paragraph, namely being an important topic in the field and a timely paper. As part of this introductory comment, the Reviewer has identified three valid concerns that we had not addressed sufficiently clearly in our initial submission:

- (a) **Major concern 1** – Generality in the form of our proposed transfer learning strategies being applicable to varied GNN architectures and being agnostic to the readout function.
- (b) **Major concern 2** – That the transfer learning strategies do not outperform the direct inclusion of low-fidelity labels in certain cases.
- (c) **Major concern 3** – That the transfer learning strategies can be detrimental to performance in certain cases.

We will address each concern individually:

- (a) **Major concern 1** – Generality in the form of our proposed transfer learning strategies

We appreciate this concern and we believe that generality and wide applicability are important aspects of a transfer learning framework of this scale. With this in mind, we would like to make several clarifications, which have also been incorporated in our revised manuscript.

Firstly, we would like to emphasise that all of the proposed transfer learning strategies are agnostic to the GNN architecture, including the readout function. Fundamentally, our framework encompasses strategies of transfer learning by **(1) embeddings**, **(2) predictions**, or **(3) supervised pre-training and fine-tuning**. These techniques are applicable to any neural network that generates molecule-level representations. For example, we will later detail newly-added experiments where SchNet is used as the underlying model instead of typical GNNs.

For a fair and extensive empirical evaluation, for every experiment where we use an adaptive readout we provide an equivalent experiment where only the readout function is changed. It is apparent that sum readout function is competitive in many tasks, particularly quantum tasks such as total, atomic, and formation energies, as well as HOMO and LUMO energies and the rotational constants. This is expected since the sum readout is naturally the right approximation for properties like the total energy, which can be viewed physically as the sum of bond formation energies. The fact that the sum readout models perform well for certain tasks is now clear from **Figure 3B** (corresponding to low-fidelity models) and from **Figure 4B**, where transfer learning with the sum readout performs competitively with the novel adaptive readout.

However, we observe consistently better performance for adaptive readouts on the drug discovery HTS datasets, which are a completely novel application and no relevant comparisons exist in the literature. This data is unique in terms of the comprising of large scale but noisy and highly-imbalanced molecule-protein interaction data. In this context, limitations of standard readouts like sum are the most apparent and limit the ability to propagate information across the screening cascade. **Figure 3A, B** illustrates the inability to learn structure-activity relationships at this scale and complexity (compound libraries are chosen to be as diverse as possible), while **Figure 3C** shows that the sum readout representations are spread out and uninformative. The fact that widely-used GNN architectures, as well as established multi-fidelity and transfer learning frameworks do not work well for drug discovery HTS data is an important discovery and one that we believe warrants further investigation. In contrast, adaptive readouts provide consistent transfer learning benefit in both applications. Our excitement about being able to model this challenging but important and underutilised data source lead us to emphasise these results more than those for quantum mechanics, creating an impression that the transfer learning strategies are dependent on this readout function. We have revised the text to make this distinction clear.

We would like to note that the adaptive readouts used here are based on Set Transformers, which are in turn an adaptation of the famous Transformer architecture for mathematical sets. Transformers are now ubiquitous in deep learning, and although they are not as commonly encountered for GNNs, they are common architectures that are widely understood and highly optimised (e.g. the recent addition of a non-approximative attention mechanism that is linear in its execution time and memory consumption in PyTorch 2.0).

Finally, we would also like to note that the first high profile multi-fidelity learning paper for GNNs (Chen et al., [Che+21]) has used a rarely-encountered readout function (Set2Set) which can be classified as an adaptive readout since it is based on a neural network.

(b) **Major concern 2 – Effectiveness of low-fidelity labels**

The concern expressed here is that the direct inclusion of the low-fidelity labels is the optimal method in some cases. To address this, it is important to highlight several different points:

- i. Supplying a model with corresponding low-fidelity labels is, in our opinion, a sensible and strong baseline for multi-fidelity learning. Since the low-fidelity label is intended to measure the same property as the high-fidelity label (albeit in a more approximate way), it carries direct signal about the label of interest. It is not surprising that the inclusion of this signal is strongly beneficial in many cases. However, this approach, while simple and effective in my cases, has a key drawback – it is restricted to the transductive setting, meaning in the case of HTS data every potential molecule for which we would like to obtain predictions must be synthesised or purchased and tested in the low-fidelity assay before a high-fidelity prediction is made. This is a massive practical drawback and we have expanded our discussion to frame this more clearly. Methods that address the *inductive* setting, without the need for experimental label collection, are critically important in real-world drug discovery. In the QM context, the requirement is to conduct a simulation at the low-fidelity settings, which while likely easier than physical experiments, may still incur substantial computational cost.
- ii. The direct inclusion of the low-fidelity labels is, in fact, not optimal in most cases. Indeed, as our summary table in **Figure 4A** shows, out of the total of 51 transductive experiments, the labels are preferable only in 10 instances. In other words, the alternative strategies proposed here are the optimal choice in over 80% of cases. Furthermore, the labels are preferable in only 2 out of 16 AstraZeneca drug discovery experiments, 4 out of the 23

PubChem experiments, and 4 out of 12 quantum properties (QMugs).

The quantum properties where labels are preferable are related to molecular dipoles, properties dependent on molecular geometry that are known to be difficult for 2D methods to predict [SUG21; Par22]. In this case, while we could still show benefit for other transfer learning methods, we could not identify a 2D graph based method that was superior to direct label inclusion. In the 6 drug discovery cases where direct label augmentation was better than other approaches (**Supplementary Information 15** and **Supplementary Table 4**), we can identify two distinct modes: these cases either have very high correlation between fidelities, implying the low fidelity label is nearly as good as the high fidelity label, or in two cases (**Figures 3A, B**), the low fidelity models are not able to learn as well as most others, leading to poor performance gain from model-based transfer learning strategies.

- iii. Despite its apparent simplicity, it is not trivial to define a transfer learning strategy that successfully leverages the low-fidelity labels. For example, the work of Chen et al. [Che+21] operates in a transductive setting and uses a custom message passing algorithm to learn from low- and high-fidelity labels during the same training epoch (more details in our new **Related work** section). However, it is ineffective for drug discovery tasks and cannot model such data, even when using just 2 fidelities (newly added **Results, Figure 6C**). This lack of performance is discussed in the **Introduction** and **Results, Section 4.5**.

Within our work, we evaluate several different transfer learning strategies with the ultimate goal of improving real-world, computational and experimental molecular workflows. Transfer learning by the direct inclusion of the low-fidelity label is a valid strategy and it performs well for many tasks, including drug discovery and quantum mechanics. We do not claim that any particular strategy is optimal all the time – in fact, our extensive evaluation justifies the use of multiple strategies for a given project, since more than one strategy can be highly performant.

(c) **Major concern 3 – Possibly detrimental performance due to transfer learning**

We understand the importance of discussing any risks associated with transfer learning and we agree with the Reviewer that this was insufficiently discussed in our initial submission. We believe that this concern can be fully resolved by considering the following points:

- i. We evaluate a relatively large number of transfer learning strategies for each multi-fidelity dataset, including ‘mirror’ experiments where the only difference is the readout function of the low-fidelity model. This enables a fair comparison with existing techniques, an aspect that was discussed at length in the response to **Major concern 1** above.

We do not claim that all transfer learning strategies achieve the same performance uplifts, and indeed, our extensive evaluation allowed us to identify the bottleneck of standard readout functions for transfer learning. From our results, particularly **Figure 4B, Figures 5B, C, Figures 6A, B, C**, and the related supplementary figures, it can be observed that transfer learning by embeddings or predictions of a low-fidelity model with the sum readout generally does not improve performance (i.e. they do not add or remove anything), while the hybrid label strategy and the supervised pre-training/fine-tuning strategy based on sum readouts can be severely detrimental.

These results are presented on purpose as they highlight the difficulty of the drug discovery tasks and the need for more powerful methods. In the initial and updated manuscript, we have focused on this aspect as part of the case study in **Results, Section 4.3** (and **Figure 5C** specifically), where we showed that models based on the sum readout learn nonsensical structure-activity relationships.

However, we acknowledged that models based on the sum readout can be effective for certain quantum tasks, especially where it makes physical sense, for example for the total, atomic, and formation energies (**Figure 4B** and **Supplementary Figure 7**). The strong results of standard readout operators in these scenarios demonstrate that (1) our evaluation is sound and (2) the method is not inherently wrong or inadequate; instead, it is unsuitable for the novel drug discovery tasks presented here, and outperformed for the quantum tasks.

- ii. We are happy to report the remarkable result that for all 51 transductive experiments, which act as our main performance benchmark, there was at least one transfer learning strategy that led to prediction performance uplifts. This means that we can always do better than just a high-fidelity model (with no low-fidelity inputs), which is unfortunately the standard approach. This is a remarkable result for multi-fidelity research and motivates the generation and modelling of such types of molecular data.

We realise that ideally, there would be a single, dominant strategy that should always be optimal. Instead, our analysis covered a wide range of assay types, screening technologies, concentrations, scoring metrics, protein targets, and experimental protocols (drug discovery) and quantum properties of different natures, and revealed that different strategies can be optimal depending on the dataset. We have also identified a hierarchy: for drug discovery, the novel supervised pre-training and fine-tuning workflow is often optimal, followed by transferring embeddings (generated by adaptive readouts). For quantum tasks, this is reversed. The other strategies are preferable much less often. These conclusions are summarised in **Figure 4A**.

- iii. It is important to emphasise that the potential of transfer learning is determined by the quality of the low-fidelity data. We have made an effort to quantify this relationship by linking the performance uplift ('% MAE decrease') with the linear correlation (Pearson's r) between the low- and high-fidelities for the compounds which have both. We performed this analysis for high fidelity models augmented with embeddings generated by a low-fidelity model with an adaptive readout.

The results are plotted in the new **Figure 5A**, which highlights moderate-to-strong, statistically significant relationships between the low- and high-fidelity correlation and the performance uplifts for AstraZeneca and PubChem drug discovery models, as well as for QMugs. In most cases, the lowest uplifts are achieved for very low correlations (≤ 0.2), with only a few exceptions to the rule. This indicates extremely noisy data, and potential experimental issues or a difficult protein to assay. In such cases, it is difficult to expect transfer learning to be successful. At the same time and as discussed before, the embeddings strategy is not always optimal, meaning that the other strategies can perform better for such datasets. We have also listed the fidelity correlations for the three fidelities encountered in the QM7b dataset (the new **Figure 6B**).

2. *'It seems for the drug dataset used, the GNN model used with sum pooling is more or less completely ineffective for the prediction of properties. From that perspective, demonstrating the effectiveness of transfer learning may get lost in the noise, so to speak. Also, much of the benefit of the transfer learning techniques highlighted really only occur with the custom adaptive pooling function, which is not widely used. As it stands, it is hard for an observer to evaluate if this work demonstrates the effectiveness of transfer learning, especially as most GNN models have some kind of sum or mean pooling.'*

Response

This is a slightly more in-depth comment touching on the same concerns as the introductory paragraph by the Reviewer, which is covered in detail as part of our response to comment 1 of Reviewer 1, particularly in the **Major concern 1** section.

In addition, we would like to note that we still use standard readouts (sum) for the high-fidelity models, which are typically small-sample (under 10,000 data points). The sum readout has also been used for all high-fidelity models (all training set sizes) in **Figure 6A**. The adaptive readouts are based on Transformers, and we currently apply them only to the large low-fidelity datasets. This workflow is detailed in **Supplementary Information 5.2**. In the updated manuscript, we also emphasise that transfer learning from sum readout models can be effective for the right tasks (e.g. the total, atomic, and formation energies from QMugs); however, it is generally ineffective for drug discovery tasks. This is a novel contribution as this type of data has not been studied with GNNs before.

Regarding the adoption of less common readouts, it is worth noting that the first and most notable high profile multi-fidelity learning paper for GNNs (Chen et al., [Che+21]) has used the Set2Set readout function, which can be classified as an adaptive readout and it is not in widespread use.

Unfortunately, the authors have not provided comparisons of their framework instantiated with standard readouts, as we do here.

3. *'It also appears using the low-fidelity label directly beats most transfer learning methods handily, especially for the QM9 type dataset in supplementary Figure 7. More discussion is warranted here, and why the fancier and more complex approaches cannot outperform a simple and direct baseline.'*

Response

This is also a slightly more in-depth comment revisiting **Major concern 2**, which has been addressed in full above. Direct label use is the best method in only a very small number of tasks.

Furthermore, we have completely redone the QMugs experiments after identifying a data filtering problem. QMugs is a publicly available high-quality dataset and we did not initially apply the same stringent data filtering steps as for the drug discovery data. However, in order to address this concern, we have investigated the dataset more closely and have discovered outlier values that were severely and negatively affecting the ability of models to learn. We have thus applied filtering measures similar to our drug discovery data, which have removed around 1% of the total 657K compounds (**Methods, Section 3.7**). For our initial version, we have also used standard GNNs for QMugs as opposed to our supervised variational architecture that is in use for the drug discovery data. This was motivated by potentially easier comparisons with existing work. However, we have now updated our quantum workflow to use the same architecture, which provides stronger performance and a more meaningful, apples-to-apples comparison with the drug discovery models.

4. *'In more than a few cases, transfer learning hurts the performance of the models significantly. This is a very concerning phenomenon as one should expect that with the additional effort of procuring low fidelity labels and additional compute for training, that at the very least this should not harm the performance. This is perhaps not discussed as much as it should be, as it represents a major potential risk in transfer learning, and it does not seem any of the proposed transfer learning methods can reliably avoid this risk.'*

Response

This is also a slightly more in-depth comment revisiting **Major concern 3**, which has been covered in detail above. To directly address the risk concern, we would like to give the example of the augmentation by neural embeddings strategy (i.e. embeddings outputted by a low-fidelity model trained with adaptive readouts), which is the second-best augmentation strategy for drug discovery data. Out of the total of 51 transductive experiments that act as our main performance benchmark, this strategy led to negative 'uplifts' in only 4 cases. By investigating each case, we have observed that they correspond to the datasets with the lowest correlations between the low- and high-fidelities. This relationship is presented graphically in the new **Figure 5A**, and the correlations for all multi-fidelity datasets are listed in **Supplementary Table 1** and **Supplementary Table 2**.

To be precise, the correlation for AID1431-873 is 0.08 (the lowest in our collection), for AID504313-2732 it is -0.09, and for AZ-DR-R13+AZ-SD11 it is -0.19, the second lowest correlation for the AstraZeneca datasets. For the other dataset, AID1259418-1259416, the correlation is slightly higher (-0.37), but the size of the low-fidelity dataset is the smallest in our collection (59,447). In these cases, it is difficult to expect transfer learning to be effective, since the measurements are either extremely noisy or essentially random as far as the high-fidelity labels are concerned. Nonetheless, even in these extreme cases the performance degradation is less than 5% on average. Moreover, the other strategies, in particular the novel supervised pre-training and fine-tuning workflow, can have positive uplifts even for these datasets, as can be seen from **Supplementary Information 6** and **Supplementary Information 10**.

5. *'There are concerns regarding the generalisability of this approach and the conclusions in this paper. The work is limited to a set of simple GNN architectures and a non-conventional pooling function. While the authors claim that different model architectures did not have a significant impact on the conclusions in the supplementary info., no convincing evidence is shown to support this argument. While understandably this involves significant additional compute for experiments, the current examples may not be entirely convincing to readers.'*

Response

We understand and appreciate the concern of an underrepresentation of machine learning algorithms in our work. We have focused on the most effective architectures and it is not feasible to examine all alternatives as our evaluation is already extensive (as also noted by Reviewer 3) and covers a large amount of space. Nonetheless, we list below the different algorithms and architectures that we tested in the revised manuscript:

- i. All the presented transfer learning strategies with the exception of the supervised pre-training and fine-tuning workflows can be used with any machine learning algorithm that uses an internal vector representation of molecules. In particular, it is an interesting and useful result that the embeddings generated by low-fidelity models with adaptive readouts can be used in downstream tasks by classical algorithms such as random forests and support vector machines. The evaluation of such models is covered in **Supplementary Information 7, 8, 11, 12**.
- ii. For this revised version we have added a series of experiments covering transfer learning for datasets with more than 2 fidelities. As a further step towards generality, we have demonstrated updated versions of our strategies for quantum tasks modelled using SchNet [Sch+17; Sch+18], a widely used neural network architecture that works directly at the level of 3D atomic coordinates. Separately [But+23], we have studied the applicability of adaptive readouts (without transfer learning) for a wide range of quantum properties using SchNet and DimeNet++ [GGG20], a leading architecture in the same family as SchNet. It is thus possible to conclude that the presented techniques apply to this family of widely used networks.
- iii. Formally, we have recently introduced adaptive readouts and extensively studied their performance across a wide range of GNN message passing algorithms [But+22a], including graph operators such as GCN, GIN, GAT, GATv2, and PNA. Although we did not consider transfer learning in the NeurIPS paper, we did experiment with 3 of the AstraZeneca drug discovery datasets and found that the training characteristics were similar for all graph operators and readout functions (including sum, mean, and maximum) in an experiment that is similar to what is covered in **Results, Section 4.1 and Figure 3**.

Furthermore, for previous versions of our manuscript, which are available online as pre-prints [But+22b], we have focused only on the drug discovery data and experimented with three types of convolutional operators: GCN, GIN, and PNA for all low-fidelity and high-fidelity models, as well as different types of standard readout functions: sum, mean, and maximum. However, the performance differences were small enough to not justify the computational resource utilisation, the time, and the increased manuscript length.

To summarise, we have chosen to prioritise architectures that enable us to devise several experiments relevant to our goals, without the burden of training variations based on graph operators or readout functions that have been established before to not lead to significant gains. The choices made in our manuscript should be considered in the context of our previous and related work, and we realise that this aspect has not been clarified initially.

6. *'Some discussion is perhaps warranted on unsupervised or self-supervised pre-training, which can provide performance benefits with no low-fidelity labels. Certainly, in some cases where the supervised pre-training benefits are minor or detrimental, one may wonder if the unsupervised approach may be more effective.'*

Response

We appreciate these suggestions and we have indeed considered such approaches, although we had not discussed them appropriately in our initial submission. First of all, we have now addressed the concerns that the Reviewer had regarding *'benefits that are minor or detrimental'* as part of our response to comment 1, in particular **Major concern 2** and **Major concern 3**. Secondly, one of the motivations for our work is making the best possible use of vast collections of **labelled** low-fidelity measurements in drug discovery, which have not been studied previously and are not used in real-world projects, in neither academia nor industry.

Nonetheless, we agree that the suggested approaches warrant a discussion:

- i. Regarding unsupervised training, there are three main reasons why we did not pursue this direction. Firstly, as stated before, completely unsupervised learning would not be the most efficient use of large collections of labelled data. Secondly, molecular representations generated through unsupervised learning are notoriously lacking in predictive utility for downstream tasks [Góm+18]. Instead, a supervised or ‘guided’ variational architecture, as we used, can learn a continuous representation of molecules where the latent space is guided by experimental measurements. This leads to a structured latent space that is specific to each drug discovery project, instead of a generic latent space generated through unsupervised learning. A supervised variational architecture is also more flexible and can be generalised in several directions, as covered in **Discussion**, for example by being used generatively or in a multi-task fashion for multiple drug discovery projects at the same time.

Thirdly, even a supervised variational architecture using standard GNNs (i.e. with the sum readout) fails to learn the structure-activity relationships of the low-fidelity dataset, as illustrated most prominently in **Figure 3C**. It is thus unlikely that unsupervised learning, using even less information, would fare better. Moreover, since the loss functions used within the variational graph autoencoder operate at the level of node representations, adaptive readouts cannot be used in an unsupervised setting.

- ii. Regarding self-supervised training, we agree that it could be a promising direction, particularly in the form of contrastive learning. However, such a decision comes with the burden of a vastly different experimental set-up. For example, contrastive learning is typically applied for classification tasks, while here we focus on regression as it is directly relevant to practitioners. While 2 datasets in our collection are formulated for classification (i.e. the pIC50 labels are not available), this is not generally the setting encountered in HTS or quantum simulations.
- iii. While work that focuses on contrastive regression exists for computer vision [Zha+23], this is a relatively novel direction and it is not obvious that it will translate well to GNNs and molecules. At the same time, recent work exists which addresses contrastive learning for molecules [Wan+22]. However, this requires bespoke training techniques such as atom masking, bond deletion, and subgraph removal, and the proposed contrastive learning method did not convincingly outperform relatively simple baselines such as directed message passing.

Overall, we believe that self-supervised learning is an interesting direction for the kinds of drug discovery and quantum multi-fidelity data presented here. However, such an exploration requires specialised techniques and an extensive evaluation that would not fit well with our current analysis, scope, and manuscript length.

Minor issues

1. *‘Figure 1 and 2 could be more illustrative for more general audiences and currently difficult to follow. It would also help if there was a figure or subfigure summarizing the 9 different methods in Figure 4 somewhere.’*

Response

We understand that Figures 1 and 2 are complex and convey a large amount of information. We feel that this situation is difficult to avoid given the large number of approaches considered here, but we have revised these figures to improve readability. In **Figure 1**, we have clearly partitioned the traditional workflow (top) and a high-level multi-fidelity modelling workflow (bottom) and their interaction through clearly delimited boxes corresponding to essential concepts such as datasets or machine learning models. The flow of information is illustrated through arrows and every step is labelled. The attached caption covers the workflow and provides further details.

Figure 2 follows the same design philosophy as **Figure 1** and we have now labelled the different transfer learning strategies, which are also listed as part of a new legend. These steps correspond to the experiments listed in the beginning of **Results**, such that it is now easier to correlate the figures to the main content and results. We have also updated the caption to reflect these changes.

For both figures, we have made slight modifications to now represent the novel supervised pre-training and fine-tuning workflow that was previously not represented. We are open to suggestions that further clarify and improve our illustrations.

2. *'Dataset mean should be provided for the QMugs dataset in Figure 3 for consistency.'*

Response

We have included the dataset mean results for QMugs in the revised version. Moreover, for a simpler and easier to understand presentation, we have normalised the MAEs using maximum absolute scaling, such that they can be visualised in the [0, 1] range without compromising the relative relationship of the methods.

3. *'Figure labels should be consistent and more clearly mentioned in the text. I.e. in Figure 4, is tune readout in 4A the same as tune vgae in 4b? If so, they should use the same name. Section 3 and more specifically 3.2 should use or mention the Figure 4a label names, or else it is difficult to track which method is which.'*

Response

We acknowledge that the strategy names can be difficult to follow, an issue also pointed out by Reviewer 3, and this is partially a result of the limited space that is available to present all of the different strategies. The particular issue of 'tune VGAE' and 'tune readout' stems from the technical differences of the supervised pre-training and fine-tuning workflow when applied using sum or adaptive readouts. More specifically, when using the sum readout, it is only possible to pre-train and fine-tune the graph layers since the readout function is fixed and non-learnable. In contrast, for adaptive readouts we freeze the graph layers and only fine-tune the neural network of the adaptive readout. In other words, for the sum readout the VGAE layers are fine-tuned, as it is the only possible choice, while for adaptive readouts the VGAE itself is frozen and only the readout is fine-tuned, hence the names used in **Figure 4**. When using adaptive readouts, it is also possible to fine-tune the VGAE component, irrespective of fine-tuning the adaptive readout, which would correspond to the 'tune VGAE' name.

For the revised manuscript, we have updated the **Methods** section corresponding to the formal description of the transfer learning strategies (**Section 3.5**), and have introduced a new section dedicated to the pre-training and fine-tuning workflows (**Section 3.6**). We have also completely dropped the term 'weight transfer' and we use the alternative 'pre-training and fine-tuning' instead. The short versions used in the figures in **Results** are now also in complete agreement with the description in **Methods**.

4. *'The % MAE decrease in Figure 4 is confusing - why not use % improvement like in Figure 5?'*

Response

This distinction is made since in **Figure 4B** the only metric that is discussed is the MAE. However, in **Figure 5B**, we report results in terms of both MAE and R^2 . Since lower is better for MAE and higher is better for R^2 , we must use a generic name for the graphs, such as '% improvement', although we would prefer the more precise '% MAE decrease' or '% R^2 increase'.

5. *'Not clear what "no low-fidelity set" means in Figure 5.'*

Response

The term is a shortened way to refer to a high-fidelity dataset that was generated later in the drug discovery campaign (after the initial screening cascade/funnel), with the important characteristic that such a dataset does not have associated low-fidelity measurements, thus the term 'no low-fidelity set'. The presentation of this type of data is handled in the first paragraph of **Results**, **Section 4.3**, and we have now clarified the meaning of the term in both the main text and the figure caption for the revised manuscript.

6. *'The switching between high fidelity and low fidelity with single dose and dose response labels are confusing.'*

Response

For the revised manuscript, we no longer use the drug discovery-specific terms and we use the same terminology across the entire paper.

Reviewer 2

1. *'This is a well written paper, which addresses an emerging area in informatics, namely how can one better leverage all of the information which is present for a particular task, rather than being restricted to a single fidelity of data, selected either by volume or quality. This is important as these two selection criteria are often in opposition – i.e. where there is large data volume, it is often the case that the quality (or more accurately the fidelity, since the measurements are often made well, but using techniques with a lot of associated noise or assumptions) is reduced.*

The authors treat the problem as a transfer learning task, transferring 'knowledge' (as it were) gained from a higher volume, lower fidelity, task to a model which includes information gleaned from higher fidelity data. It should be noted that as a parallel submission to this effort, the authors have made available a set of benchmark datasets, which are of great value to the community.'

Response

We would like to thank the Reviewer for appreciating our work and highlighting some of the strong attributes and contributions. This introductory paragraph recognises the importance of multi-fidelity data generation and modelling, while acknowledging the challenges and limited research in this area.

2. *'One theme that does persist through the paper, perhaps due to the authors' key areas of expertise, is somewhat of a lack of appreciation for the literature on multi-fidelity machine learning and optimization in the materials chemistry domain. Some examples which the authors might find helpful to consider contrasting and comparing their approach to could be:*

1. Yang, C.-H. et al. Multi-fidelity machine learning models for structure–property mapping of organic electronics. *Computational Materials Science* 213, 111599 (2022).

2. Patra, A. et al. A multi-fidelity information-fusion approach to machine learn and predict polymer bandgap. *Computational Materials Science* 172, 109286 (2020).

3. Meng, X. & Karniadakis, G. E. A composite neural network that learns from multi-fidelity data: Application to function approximation and inverse PDE problems. *Journal of Computational Physics* 401, 109020 (2020).

4. Chen, C., Zuo, Y., Ye, W., Li, X. & Ong, S. P. Learning properties of ordered and disordered materials from multi-fidelity data. *Nat Comput Sci* 1, 46–53 (2021).

5. Fare, C., Fenner, P., Benatan, M., Varsi, A. & Pyzer-Knapp, E. O. A multi-fidelity machine learning approach to high throughput materials screening. *npj Computational Materials* 8, 1–9 (2022).

Response

We would like to thank the Reviewer for providing a critical selection of relevant literature. To rectify the highlighted shortcoming, we have introduced a new **Related work** section after the introduction, positioning our work in the context of the existing literature. The studies suggested by the Reviewer are certainly relevant for the field of multi-fidelity and transfer learning research, and are appropriately discussed in our revised manuscript.

Moreover, we are happy to report that we have gone one step further and have added a new suite of experiments specifically to compare with the work of Chen et al. [Che+21] (reference 4 in the list above), which is one of the first and only high-profile multi-fidelity learning frameworks for GNNs. The new analysis is presented in **Results, Section 4.5** and it indicates that this existing framework performs poorly for the novel drug discovery tasks introduced in our work. This result further motivates the need for innovations in transfer learning tailored for molecular data.

3. *'It is my opinion that publication of this type of work in a journal such as Nature Communications requires that there be first of a kind novelty beyond a specific domain (such as drug discovery), or significant performance improvements with real world consequences. Without the consideration of methods beyond those applied specifically to drug discovery, it is hard for me as a reviewer to make that judgement.'*

Response

We believe that both of these criteria for novelty are satisfied. Although we will present the contributions point-wise for clarity, it is the holistic transfer learning framework that combines methodological, algorithmic, and domain-specific contributions that addresses real-world problems which were not solvable before.

The novel aspects of our work are presented in detail in the *Novelty statement* section at the beginning of this letter.

4. *'The authors tend to treat each low-fidelity data source as an augmentation to a single model. I am unsure as to how this approach would naturally deal with, for example, two different ways of measuring the same target. This is common in quantum chemistry, for example. Since quantum chemistry is one of the use cases here, I would suggest that the authors consider how to expand their work to include these kinds of task. This would imply that the authors be able to cover the following questions: (a) the effectiveness of transfer learning when there are more than two levels (e.g. high and low). A comparison of a strategy where models are built through stepwise pairing and when models are built through many-to-one transfer would be helpful. This would provide strength to the main value proposition of the paper, namely that one should use all available data, rather than a single fidelity. (b) The interplay between different fidelities or augmentations into the model. When more than two fidelities are investigated, there is much to be learned from the interactions between two non-target fidelities which can inform, for example, further data collection. An example may be to understand how electronic properties calculated at one combination of functional/approach and basis set might relate to another. (c) The response of a model in the multi-fidelity setting, where one of the lower fidelities are non-correlated (i.e. poisonous to the high fidelity model), especially in areas of configuration space. An example of this is the use of simulated data, where the simulation model may have been tuned for a subset of chemical space and has degraded performance outside of this space.'*

It should be noted that principled methods for generating synthetic fidelities based on 'anchoring' fidelities exist, with one recent example being:

Fare, C., Fenner, P. & Pyzer-Knapp, E. O. A Principled Method for the Creation of Synthetic Multi-fidelity Data Sets. Preprint at <https://doi.org/10.48550/arXiv.2208.05667> (2022).'

Response

We are pleased to report that we have now extended our manuscript with an analysis of transfer learning for more than 2 fidelities, based on a real-world quantum mechanics dataset (QM7b). While the idea of a synthetically-generated dataset is appealing, we consider that the required domain expertise, determining the right parameters for the resulting dataset, and the drawn conclusions would significantly affect the length and scope of our manuscript, which already covers a large number of strategies, experiments, and results, and was already described as intricate and occasionally difficult to follow by Reviewers 1 and 3. Furthermore, a possibly worry is that results based on simulated data might not be appreciated to the same extent as real-world data, particularly when such large collections of data already exist, but are underutilised and understudied.

Nonetheless, we have discussed this direction in our updated manuscript, including citing the mentioned work, and have taken several steps to better link our results to the correlations within the fidelities, which we will describe shortly. Moreover, in studying QM7b, we take the opportunity to demonstrate our transfer learning strategies in the context of SchNet, a notable extension compared to the standard GNNs that we had used so far.

The transfer learning strategies that involve transfer by embeddings or predictions can be trivially adapted to more than 2 fidelities by training a different model for each fidelity. Then, the outputs can be combined (by concatenation) to the internal representation of the chosen target fidelity.

The QM7b properties that we study are the HOMO and LUMO energies in a transductive scenario, which are provided at 3 levels of theory: ZINDO, PBE0, and GW, and which are also endowed with an ‘ordering’: ZINDO < PBE0 < GW. The experimental setup and the results are provided in the new **Results, Section 4.6** and the new **Figure 6B**. We also include the linear correlation between the different levels of theory (Pearson’s r). From the analysis, we can derive two important conclusions: (1) the ZINDO and PBE0 labels, both when applied in isolation or together, provide a very limited increase in predictive performance for the target GW fidelity. This indicates that one of the novel strategies must be used. (2) When using transfer learning by ZINDO and PBE0 embeddings (generated using adaptive readouts) jointly, the performance uplift is higher than for each fidelity individually. This is interesting since ZINDO is a relatively crude calculation, which is poorly correlated with GW, particularly for LUMO energy ($r = 0.5$). The fact that the addition of ZINDO information improves upon PBE0 suggests that the structure-activity relationships contained within the embeddings enable smoother training and a higher performance ceiling.

Furthermore, we have calculated the fidelity correlations for all drug discovery datasets and QMugs properties (listed in **Supplementary Table 1, 2, 3**) and linked them to the observed performance uplifts (the new **Figure 5A**). We identified moderate-to-strong, statistically significant relationships between the low- and high-fidelity correlation and the performance uplifts. Generally, the lowest uplifts are achieved for very low correlations (≤ 0.2), which can indicate extremely noisy data, and potential experimental issues or a difficult protein to assay. Such cases correspond to the example given by the Reviewer where the extra fidelity can actually be harmful. However, **Figure 5A** only captures the transfer learning by embedding strategy. As a counter-example, for the public drug discovery dataset AID873-1431 with a fidelity correlation of 0.08, the embeddings did not lead to a performance uplift, but the novel supervised pre-training and fine-tuning strategy decreased the MAE by more than 41% (**Supplementary Figure 8**). This demonstrates that the fidelity correlation is definitely not a perfect measure of the expected uplift, although it is often a good indicator.

5. *‘I found it unclear from the paper as to the requirement for complete data coverage over the low-fidelity tasks. That is, if there are multiple data sources with different ‘fidelities’ of data, does the training model need to have complete data sets to train, or can it deal with partial data across the multi-fidelity regime? I would suggest that being able to deal with multiple fidelities implies that the model should really be able to deal with datasets where there is not complete coverage of targets at all fidelities to maximise the impact of the method. It would be good for the authors to directly address this point.’*

Response

We agree with the Reviewer that this is a crucial aspect of any transfer learning framework. While we have detailed the structure of the multi-fidelity datasets in **Methods, Section 3.6**, and have listed the number of molecules for each fidelity and for all datasets in **Supplementary Tables 1 and 2**, we did not explicitly and clearly mention this in the main text. We have improved this in the revised version.

To summarise, all the transfer learning strategies presented here are agnostic to the number of molecules measured at each fidelity level and it is not required that there is complete data coverage over the low-fidelity tasks. The best and most prominent examples are the drug discovery datasets: the low-fidelity measurements correspond to primary screens which cover up to 2 million compounds, while the high-fidelity confirmatory screen generally covers less than 10,000 compounds. Since our strategies rely on training separate models for each fidelity (or supervised pre-training and fine-tuning, which still starts by pre-training on the low-fidelity data), the number of molecules within the fidelities does not matter. The QMugs and QM7b datasets do have full data coverage, i.e. all molecules are associated with measurements at every fidelity. However, even in this case we have an extensive suite of experiments where we vary the high-fidelity training set size to much smaller values, such as 8K, 25K, 50K, 100K, and 300K (**Figure 6A** and **Supplementary Figure 15**).

We think that the Reviewer will also appreciate that the updated manuscript more prominently discusses and explains the *transductive* and *inductive* settings, as these are linked to the data coverage question. We have also covered this distinction at length in this letter, particularly in the *Novelty statement* at the beginning.

6. *'In summary, I think this is an interesting paper which has some promise. Before I can recommend its acceptance and publication in a journal such as Nature Communications, I would require that the questions I have raised, namely around comparisons to existing literature from the materials chemistry domain, and the applicability of the proposed method to more than 'high/low' fidelity classification and the requirements for data completeness in the low fidelity regimes, be addressed.'*

Response

We are thankful for the insightful questions and the appreciation of our work so far. These suggestions have unmistakably led to stronger results and a more general and capable framework. Naturally, we are happy and open to receiving additional feedback and continuing to refine our work.

Reviewer 3

1. *'In current manuscript, authors described various methodologies for infusing models learned from dense low fidelity data with models learned from sparse high fidelity data to increase model performance on sparse high fidelity data. The authors tried various fusion strategies as well as different training settings (transductive vs inductive setting). The included datasets are quite extensive including both industry and open source.'*

Response

We would like to thank the Reviewer for the positive outlook on our work. In particular, the Reviewer has succinctly described one of our contributions, namely the extensive analysis covering several transfer learning strategies in both the *transductive* and *inductive* cases. We will clarify other aspects of our contributions as part of our answer to comment 2 of Reviewer 3 (the next one).

2. *'Although it is interesting to see how different fusion strategy can improve the performance comparing with model learned on sparse high fidelity data alone, I feel that both the applied fusion strategy and neural network architecture are not novel enough to be accepted for this journal.'*

Response

We appreciate that our work covers a large range of experiments and contributions and in the previous version of our manuscript these were not as clearly accentuated. We would like to start by noting that transfer learning for molecules, and especially for applications in drug discovery, is an emerging area of informatics with a considerable amount of recent interest, as pointed out by the other two Reviewers.

The novel aspects of our work are presented in detail in the *Novelty statement* section at the beginning of this letter.

3. *'The authors are quite ambitious to make a systematic survey. But, in my opinion, considering too many scenario/strategies makes the manuscript hard to follow. It is unclear to me in the end, which fusion strategy is the best one to recommend. The authors can probably consider to simplify the structure of the manuscript.'*

Response

We appreciate the concern that our evaluation might be too intricate. However, we strongly believe that such an extensive evaluation is necessary as we introduce several novel datasets, concepts, and strategies, and the burden of demonstrating their value and performance lies with us. For a fair comparison, we have included experiments for both adaptive readouts (novel) and standard readout functions (existing), thus effectively doubling the number of experiments and results. Moreover, both Reviewer 1 and 2 have identified a number of valid extensions to our previous version of the work, more specifically by adapting it to more than 2 fidelities and to more GNN architectures. We agree that the additional experiments and comparisons further motivate the generation and modelling of multi-fidelity data, thus further expanding our evaluation.

Nonetheless, we have implemented additional steps to help with the readability and presentation of our work. We have adjusted **Figures 1 and 2** to also illustrate the supervised pre-training and fine-tuning workflow unique to adaptive readouts, and labelled the experiments depicted in **Figure 2**, which are also summarised in a new legend. We have also updated the **Methods** section to be more concrete and better aligned with our notation throughout the rest of the paper, particularly for the pre-training and fine-tuning workflows that were previously not well explained. The **Results** section starts with a bullet list of each major contribution and its associated experiments, along with the involved strategies.

Regarding the best strategy concern, we had already provided a summary of the optimal strategies for each dataset group (AstraZeneca drug discovery, PubChem drug discovery, and QMugs) as part of the table in **Figure 4A**, which was updated with the latest results in the revised version

of the manuscript. Our extensive evaluation reveals that there is not a single strategy that is universally optimal. However, out of all the considered strategies, it is the novel supervised pre-training and fine-tuning strategy which is most frequently the best, followed by the novel transfer by neural embeddings strategy. The few cases where the performance between the low-fidelity labels strategy and the strategies based on adaptive readouts was very close are discussed and explained in a new section (**Supplementary Information 15** and **Supplementary Table 4**).

4. *'It seems some strategies was mentioned earlier, but it was not listed and discussed later (at least clearly matched to the methods in Figure 4). For example, "weight transfer as in Smith et al." was mentioned as a strategy for comparison. It is unclear to me which augmentation method in Figure 4, 5 is for this strategy (?).'*

Response

We acknowledge that the previous notation was not consistent and sometimes confusing. This was also pointed out in minor comment 3 of Reviewer 1 (page 9). In addition, the strategy 'weight transfer as in Smith et al.' corresponds to the 'tune VGAE' strategy from **Figure 4A, B**, with the adaptive readout counterpart 'tune readout'. Both were explained in detail as part of **Methods, Section 3.1**, after **Definition 1** in our previous iteration of the manuscript.

For the revised version, we have updated and extended the relevant **Methods** sections, in particular **Section 3.5** and the new **Section 3.6**, and we now use a consistent notation throughout the paper. We also completely dropped the 'weight transfer' term and replaced it with the more common 'pre-training and fine-tuning'. The introductory paragraph in **Results** and the 7-item list that follows also clearly outline how the experiments are organised into subsections, and which previous techniques (e.g. by Smith et al. and Chen et al.) are discussed and where.

5. *'The authors compared fused models with models without fusion. But model structure of non-fusion model is the same with those fused models. This is perhaps not surprising since more low fidelity data was implicitly included in the training on high fidelity model. What about comparing with non-fusion models using other network architecture?'*

Response

This observation most likely refers to the high-level workflow depicted in **Figure 2**. As emphasised in our revised manuscript, one of the core ideas of our general transfer learning framework is training a separate model for each fidelity. Then, we devised and evaluated strategies that enable the propagation of information across the screening cascade, by transferring knowledge from the low-fidelity model to the high-fidelity model. In this framework, it is **not** required that the low- and high-fidelity architectures are the same (or for each fidelity in the general case).

In particular, we are already using different architectures for experiments that involve low- and high-fidelity models. For low-fidelity models, we have established that adaptive readouts are crucial for unlocking transfer learning capabilities and propagating information towards small-sample high-fidelity models (**Results, Section 4.1** and **Figure 3A, B**). However, adaptive readouts are based on the famous Transformer architecture (**Methods, Section 3.3**), which is generally considered data hungry. For this reason, all high-fidelity models (which operate in a limited-data regime, typically with less than 10,000 data points) use standard readouts, which are intuitively more suitable for this kind of task. Furthermore, the low- and high-fidelity models that we train use different hyperparameters.

In summary, the low- and high-fidelity models are free to use different architectures. For simplicity, we have used our own supervised VGAE architecture as the basis for both fidelities (excepting the SchNet experiments), but have used different readout functions and hyperparameters. This modularity is one of the strengths of the proposed framework, and in the revised version we have clarified the experimental setup, which is detailed in **Supplementary Information 5.2**.

References

- [Sch+17] Kristof Schütt et al. ‘SchNet: A continuous-filter convolutional neural network for modeling quantum interactions’. In: *Advances in Neural Information Processing Systems*. Ed. by I. Guyon et al. Vol. 30. Curran Associates, Inc., 2017.
- [Góm+18] Rafael Gómez-Bombarelli et al. ‘Automatic Chemical Design Using a Data-Driven Continuous Representation of Molecules’. In: *ACS Central Science* 4.2 (2018), pp. 268–276.
- [Sch+18] K. T. Schütt et al. ‘SchNet – A deep learning architecture for molecules and materials’. In: *The Journal of Chemical Physics* 148.24 (2018), p. 241722.
- [GGG20] Johannes Gastegger, Janek Groß and Stephan Günnemann. ‘Directional Message Passing for Molecular Graphs’. In: *International Conference on Learning Representations*. 2020.
- [Hu+20] Weihua Hu et al. ‘Strategies for Pre-training Graph Neural Networks’. In: *International Conference on Learning Representations*. 2020.
- [Che+21] Chi Chen et al. ‘Learning properties of ordered and disordered materials from multi-fidelity data’. In: *Nature Computational Science* 1.1 (Jan. 2021), pp. 46–53. ISSN: 2662-8457.
- [SUG21] Kristof Schütt, Oliver Unke and Michael Gastegger. ‘Equivariant message passing for the prediction of tensorial properties and molecular spectra’. In: *Proceedings of the 38th International Conference on Machine Learning*. Ed. by Marina Meila and Tong Zhang. Vol. 139. Proceedings of Machine Learning Research. PMLR, July 2021, pp. 9377–9388.
- [But+22a] David Buterez et al. ‘Graph Neural Networks with Adaptive Readouts’. In: *Advances in Neural Information Processing Systems*. Ed. by S. Koyejo et al. Vol. 35. Curran Associates, Inc., 2022, pp. 19746–19758.
- [But+22b] David Buterez et al. ‘Multi-fidelity machine learning models for improved high-throughput screening predictions’. In: *ChemRxiv* (2022).
- [Ise+22] Clemens Isert et al. ‘QMugs, quantum mechanical properties of drug-like molecules’. In: *Scientific Data* 9.1 (June 2022), p. 273. ISSN: 2052-4463.
- [Par22] Yang Jeong Park. *Edge Direction-invariant Graph Neural Networks for Molecular Dipole Moments Prediction*. 2022. arXiv: 2206.12867 [cs.LG].
- [Wan+22] Yuyang Wang et al. ‘Molecular contrastive learning of representations via graph neural networks’. In: *Nature Machine Intelligence* 4.3 (Mar. 2022), pp. 279–287. ISSN: 2522-5839.
- [But+23] David Buterez et al. ‘Modelling local and general quantum mechanical properties with attention-based pooling’. In: *ChemRxiv* (2023).
- [Zha+23] Kaiwen Zha et al. *Supervised Contrastive Regression*. 2023. URL: https://openreview.net/forum?id=_QZ1je4dZPu.

REVIEWERS' COMMENTS

Reviewer #1 (Remarks to the Author):

The authors should be commended for their thorough responses to all the points raised in the previous review. The manuscript has been extensively revised with additional clarification and experiments. In its current state this is a very thorough look into multi fidelity transfer learning for molecular data on a number of established as well as novel datasets. The manuscript is publishable in its current state.

Reviewer #2 (Remarks to the Author):

For clarity, I am attaching a response to your letter. I think the manuscript is significantly improved, and commend the authors on their detailed and thoughtful improvements.

Novelty statement

First of all, we believe that it is just as important to consider the algorithmic contributions which are foundational to graph representation learning. We empirically demonstrated the limited potential of standard GNN architectures as part of different transfer learning strategies and identified the bottleneck: the readout function. We have extensively discussed the interplay between the different readout functions in our response to Reviewer 1's comments (below). Most importantly, our proposed adaptive readouts unlock a powerful and novel transfer learning strategy for GNNs: supervised pre-training and fine-tuning of the readout function, which is similar in scope and usability to the prevalent pre-training schemes used in Large Language Models, but has eluded GNNs so far. Pre-training and fine-tuning GNNs is a notoriously difficult problem [Hu+20] that has been known for a few years, particularly for molecular tasks, which also explains the relatively scarce literature for this type of transfer learning for GNNs. We also introduce a supervised VGAE model that leverages our adaptive readout functions. While being constructed of previously-developed components, it is a novel model architecture for multifidelity learning that is observed to offer substantial benefits for transfer learning.

I think that this point is well made. Innovative combination and use of known ideas does not preclude novelty in the outcome, although it does mean that the novelty must be more clearly and robustly articulated.

Our contribution is not limited in scope to a single domain. While we have focused on drug discovery, generality was one of the aims for our framework. We extensively cover multi-fidelity learning for the quantum mechanics dataset QMugs, the only multi-fidelity quantum dataset of comparable scale to our drug discovery data [Ise+22]. Our analysis spans a number of quantum properties of different types, which are tackled both *transductively* and *inductively*. Remarkably, although transfer learning is more established in the field of quantum machine learning, we are still able to outperform existing strategies when using the strategies developed here. Secondly, in our revised evaluation we extend our strategies to cover cases with more than 2 fidelities. We take this opportunity to also demonstrate that transfer learning as presented here also works for SchNet [Sch+17; Sch+18], and thus for the widely-used family of deep learning models operating at the level of 3D atomic coordinates and physically-relevant quantities like bond distances and angles. In summary, our proposed methodologies show benefit across a large number of molecular properties, settings and model types.

Again, the point is well made. I would caution about expressing generality in too strong a term here, as (through no fault of the authors) the scale and availability of challenging data and tasks in (e.g.) the materials domain does preclude as stern a test as in the drug discovery domain. For example, the QM-7 (etc) datasets lack the complexity and diversity of similarly sized drug discovery datasets due to the manner of their construction. Additionally, as the authors note later, simulated datasets are often not as challenging for ML as 'real world experimental' datasets, since we know that there exists mathematical underpinnings, which are most often noiseless in origin. Finally, to my knowledge, the drug discovery domain is the only domain which utilises in its published datasets, the concept of 'decoy' molecules – which are designed to introduce activity cliffs into data-derived QSAR type models. To be clear, this is not to say that the proposed methods would not work in other domains, or that the authors have not made best efforts to demonstrate transferability, but that strong claims of generality are probably best demonstrated through wider leverage by the community, and I would not want a good paper to fall prey to the perceived requirement for 'hype' which might preclude or inhibit the publication of later works which could strengthen cross-domain utilisation.

We show significant improvements that we believe have substantial potential impacts on the real world. Within drug discovery, our approach is a completely novel application and we utilise public as well as commercial HTS data from historical drug discovery campaigns. This data has not, to our knowledge, been investigated in a multi-fidelity setting, despite being a core technology in the pharmaceutical industry. As such, there are potential direct benefits for real-world drug discovery projects which were previously unexplored. We would also like to explicitly distinguish between the *transductive* (high- and low-fidelity data available for all points) and the more challenging *inductive* settings (with only partial low-fidelity coverage). The transductive setting acts as an excellent benchmark for molecular transfer learning, as having both fidelities enables the comparison with an 'ideal' baseline where the actual lowfidelity label is available, and it is the approach taken in previous multi-fidelity studies, including the references suggested by the Reviewer. However, the more challenging *inductive* setting has generally been neglected, despite being of direct practical relevance: when low-fidelity experimental labels are required for a method, it cannot be used to assess which molecules to make or purchase, since these costs and latency will outstrip any difference in the cost of different assays. This scenario is virtually always seen in drug discovery by

HTS projects, since the primary (low-fidelity) screening stage is followed by an optimisation where new molecules are designed, made and tested.

In this context, we emphasise that our extensive evaluation reveals that no previous multi-fidelity or transfer learning techniques works for the drug discovery by HTS datasets. In contrast, we are able to report uplifts in predictive performance using our new methods between 20% and 80% depending on the dataset, and severalfold in the best cases, for both *transductive* and *inductive* settings. Our most striking result is that transfer learning as presented here can improve the accuracy of predictive models by up to 8 times while using an order of magnitude less high-fidelity training data (**Results, Section 4.4** and **Figure 6A**). In the case of drug discovery, transfer learning by embeddings learnt in an inductive setting led to the same performance as the low-fidelity labels themselves (usable only in the transductive setting). This is a remarkable result as the low-fidelity information must be extrapolated from models that have not encountered the molecules before, indicating strong generalisation performance.

While we are not able to disclose impacts of these methods on active commercial projects, this would constitute an unreasonable standard for impact that very few academic papers would meet. Our ultimate vision for the impact of this work in the real-world is to integrate transfer learning into wet-lab experimental protocols, allowing new, promising compounds to be investigated (which would otherwise be missed by traditional methods), cost savings by a more focused and less wasteful experimental validation, and allowing new capabilities that transcend single projects, for example by training a 'universal' *in silico* low-fidelity simulator trained on hundreds of project assays, or adapting our variational architectures to generative uses. Several of these ideas are being trialled internally and we believe that we have made the first big step forward through the work presented here.

I would like to echo the authors words here – it would be unreasonable to expect a commercial entity to include application details for active commercial projects.

Reviewer 2

1. *This is a well written paper, which addresses an emerging area in informatics, namely how can one better leverage all of the information which is present for a particular task, rather than being restricted to a single fidelity of data, selected either by volume or quality. This is important as these two selection criteria are often in opposition – i.e. where there is large data volume, it is often the case that the quality (or more accurately the fidelity, since the measurements are often made well, but using techniques with a lot of associated noise or assumptions) is reduced.*

The authors treat the problem as a transfer learning task, transferring 'knowledge' (as it were) gained from a higher volume, lower fidelity, task to a model which includes information gleaned from higher fidelity data. It should be noted that as a parallel submission to this effort, the authors have made available a set of benchmark datasets, which are of great value to the community. '

Response

We would like to thank the Reviewer for appreciating our work and highlighting some of the strong attributes and contributions. This introductory paragraph recognises the importance of multi-fidelity data generation and modelling, while acknowledging the challenges and limited research in this area.

2. *'One theme that does persist through the paper, perhaps due to the authors' key areas of expertise, is somewhat of a lack of appreciation for the literature on multi-fidelity machine learning and optimization in the materials chemistry domain. Some examples which the authors might find helpful to consider contrasting and comparing their approach to could be:*

1. Yang, C.-H. et al. Multi-fidelity machine learning models for structure–property mapping of organicelectronics. *Computational Materials Science* 213, 111599 (2022).

2. Patra, A. et al. A multi-fidelity information-fusion approach to machine learn and predict polymerbandgap. *Computational Materials Science* 172, 109286 (2020).

3. Meng, X. & Karniadakis, G. E. A composite neural network that learns from multi-fidelity data: Application to function approximation and inverse PDE problems. *Journal of Computational Physics* 401, 109020 (2020).

4. Chen, C., Zuo, Y., Ye, W., Li, X. & Ong, S. P. Learning properties of ordered and disordered materials from multi-fidelity data. *Nat Comput Sci* 1, 46–53 (2021).

5. Fare, C., Fenner, P., Benatan, M., Varsi, A. & Pyzer-Knapp, E. O. A multi-fidelity machine learning approach to high throughput materials screening. *npj Computational Materials* 8, 1–9 (2022). ’

Response

We would like to thank the Reviewer for providing a critical selection of relevant literature. To rectify the highlighted shortcoming, we have introduced a new **Related work** section after the introduction, positioning our work in the context of the existing literature. The studies suggested by the Reviewer are certainly relevant for the field of multi-fidelity and transfer learning research, and are appropriately discussed in our revised manuscript.

From reading this updated manuscript, I have now come to the realisation that the major application the authors see is the utilisation of multiple data fidelities to build better high-fidelity models. In the previous version, this was blurred with the task of using those models for discovery. This section significantly improves the manuscript, as it places the work in a particular context (i.e. pre-existence of large amounts of especially low fidelity data). This places it as distinct from pure active learning type approaches, such as Fare et al, where scaling is less of an issue as the *raison d’etre* is to reduce dataset size to a minimum. One point I would make about this section is that there is the implication that Bayesian optimization does not utilise the uncertainty quantification aspect of Gaussian processes. What I think that the authors mean by their statement is that this paper does not study the quality / richness of Gaussian processes for this task. The wording as is could lead to a causal reader misrepresenting the authors’ opinion.

Moreover, we are happy to report that we have gone one step further and have added a new suite of experiments specifically to compare with the work of Chen et al. [Che+21] (reference 4 in the list above), which is one of the first and only high-profile multi-fidelity learning frameworks for GNNs. The new analysis is presented in **Results, Section 4.5** and it indicates that this existing framework performs poorly for the novel drug discovery tasks introduced in our work. This result further motivates the need for innovations in transfer learning tailored for molecular data.

I commend the authors for adding this comparison. It is also clear in the paper why this method was chosen to benchmark.

3. *‘It is my opinion that publication of this type of work in a journal such as Nature Communications requires that there be first of a kind novelty beyond a specific domain (such as drug discovery), or significant performance improvements with real world consequences. Without the consideration of methods beyond those applied specifically to drug discovery, it is hard for me as a reviewer to make that judgement.’*

Response

We believe that both of these criteria for novelty are satisfied. Although we will present the contributions point-wise for clarity, it is the holistic transfer learning framework that combines methodological, algorithmic, and domain-specific contributions that addresses real-world problems which were not solvable before.

The novel aspects of our work are presented in detail in the *Novelty statement* section at the beginning of this letter.

Given the improved context sections, and the best efforts of the authors to include comparisons, I have changed my opinion here. I made these comments in the understanding that the 'discovery' aspect of the work was as key to the novelty as the building of the models. With the improved paper contents and structure, the novelty is now clearer, and I think the bar has been passed.

4. *The authors tend to treat each low-fidelity data source as an augmentation to a single model. I am unsure as to how this approach would naturally deal with, for example, two different ways of measuring the same target. This is common in quantum chemistry, for example. Since quantum chemistry is one of the use cases here, I would suggest that the authors consider how to expand their work to include these kinds of task. This would imply that the authors be able to cover the following questions: (a) the effectiveness of transfer learning when there are more than two levels (e.g. high and low). A comparison of a strategy where models are built through stepwise pairing and when models are built through many-to-one transfer would be helpful. This would provide strength to the main value proposition of the paper, namely that one should use all available data, rather than a single fidelity. (b) The interplay between different fidelities or augmentations into the model. When more than two fidelities are investigated, there is much to be learned from the interactions between two non-target fidelities which can inform, for example, further data collection. An example may be to understand how electronic properties calculated at one combination of functional/approach and basis set might relate to another. (c) The response of a model in the multi-fidelity setting, where one of the lower fidelities are non-correlated (i.e. poisonous to the high fidelity model), especially in areas of configuration space. An example of this is the use of simulated data, where the simulation model may have been tuned for a subset of chemical space and has degraded performance outside of this space.*

It should be noted that principled methods for generating synthetic fidelities based on 'anchoring' fidelities exist, with one recent example being:

Fare, C., Fenner, P. & Pyzer-Knapp, E. O. A Principled Method for the Creation of Synthetic Multi-fidelity Data Sets. Preprint at <https://doi.org/10.48550/arXiv.2208.05667> (2022). '

Response

We are pleased to report that we have now extended our manuscript with an analysis of transfer learning for more than 2 fidelities, based on a real-world quantum mechanics dataset (QM7b). While the idea of a synthetically-generated dataset is appealing, we consider that the required domain expertise, determining the right parameters for the resulting dataset, and the drawn conclusions would significantly affect the length and scope of our manuscript, which already covers a large number of strategies, experiments, and results, and was already described as intricate and occasionally difficult to follow by Reviewers 1 and 3. Furthermore, a possibly worry is that results based on simulated data might not be appreciated to the same extent as real-world data, particularly when such large collections of data already exist, but are underutilised and understudied.

Nonetheless, we have discussed this direction in our updated manuscript, including citing the mentioned work, and have taken several steps to better link our results to the correlations within the fidelities, which we will describe shortly. Moreover, in studying QM7b, we take the opportunity to demonstrate our transfer learning strategies in the context of SchNet, a notable extension compared to the standard GNNs that we had used so far.

The transfer learning strategies that involve transfer by embeddings or predictions can be trivially adapted to more than 2 fidelities by training a different model for each fidelity. Then, the outputs can be combined (by concatenation) to the internal representation of the chosen target fidelity.

The QM7b properties that we study are the HOMO and LUMO energies in a transductive scenario, which are provided at 3 levels of theory: ZINDO, PBE0, and GW, and which are also endowed with an 'ordering': ZINDO < PBE0 < GW. The experimental setup and the results are provided in the new **Results, Section 4.6** and the new **Figure 6B**. We also include the linear correlation between the different levels of theory (Pearson's r). From the analysis, we can derive two important conclusions: **(1)** the ZINDO and PBE0 labels, both when applied in isolation or together, provide a very limited increase in predictive performance for the target GW fidelity. This indicates that one of the novel strategies must be used. **(2)** When using transfer learning by ZINDO and PBE0 embeddings (generated using adaptive readouts) jointly, the performance uplift is higher than for each fidelity individually. This is interesting since ZINDO is a relatively crude calculation, which is poorly correlated with GW, particularly for LUMO energy ($r = 0.5$). The fact that the addition of ZINDO information improves upon PBE0 suggests that the structure-activity relationships contained within the embeddings enable smoother training and a higher performance ceiling.

Furthermore, we have calculated the fidelity correlations for all drug discovery datasets and QMugs properties (listed in **Supplementary Table 1, 2, 3**) and linked them to the observed performance uplifts (the new **Figure 5A**). We identified moderate-to-strong, statistically significant relationships between the low- and high-fidelity correlation and the performance uplifts. Generally, the lowest uplifts are achieved for very low correlations (≤ 0.2), which can indicate extremely noisy data, and potential experimental issues or a difficult protein to assay. Such cases correspond to the example given by the Reviewer where the extra fidelity can actually be harmful. However, **Figure 5A** only captures the transfer learning by embedding strategy. As a counter-example, for the public drug discovery dataset AID873-1431 with a fidelity correlation of 0.08, the embeddings did not lead to a performance uplift, but the novel supervised pre-training and fine-tuning strategy decreased the MAE by more than 41% (**Supplementary Figure 8**). This demonstrates that the fidelity correlation is definitely not a perfect measure of the expected uplift, although it is often a good indicator.

This has improved the paper considerably, in my opinion. I particularly appreciate the consideration given to potentially 'poisoning' datasets. For future work, I would suggest investigating the prediction of transferability in more detail, as a strengthened indicator of utility for particular tasks could significantly impact its real-world utilisation. For this, you might find the following papers insightful:

Phys. Chem. Chem. Phys., 2020, 22, 13041-13048

<https://arxiv.org/abs/1502.02072>

J. Chem. Inf. Model., 2017, 57, 2490-2504

5. *'I found it unclear from the paper as to the requirement for complete data coverage over the low-fidelity tasks. That is, if there are multiple data sources with different 'fidelities' of data, does the training model need to have complete data sets to train, or can it deal with partial data across the multi-fidelity regime? I would suggest that being able to deal with multiple fidelities implies that the model should really be able to deal with datasets where there is not complete coverage of targets at all fidelities to maximise the impact of the method. It would be good for the authors to directly address this point.'*

Response

We agree with the Reviewer that this is a crucial aspect of any transfer learning framework. While we have detailed the structure of the multi-fidelity datasets in **Methods, Section 3.6**, and have listed the number of molecules for each fidelity and for all datasets in **Supplementary Tables 1 and 2**, we did not explicitly and clearly mention this in the main text. We have improved this in the revised version.

To summarise, all the transfer learning strategies presented here are agnostic to the number of molecules measured at each fidelity level and it is not required that there is complete data coverage over the low-

fidelity tasks. The best and most prominent examples are the drug discovery datasets: the low-fidelity measurements correspond to primary screens which cover up to 2

million compounds, while the high-fidelity confirmatory screen generally covers less than 10,000 compounds. Since our strategies rely on training separate models for each fidelity (or supervised pre-training and fine-tuning, which still starts by pre-training on the low-fidelity data), the number of molecules within the fidelities does not matter. The QMugs and QM7b datasets do have full data coverage, i.e. all molecules are associated with measurements at every fidelity. However, even in this case we have an extensive suite of experiments where we vary the high-fidelity training set size to much smaller values, such as 8K, 25K, 50K, 100K, and 300K (**Figure 6A** and **Supplementary Figure 15**).

We think that the Reviewer will also appreciate that the updated manuscript more prominently discusses and explains the *transductive* and *inductive* settings, as these are linked to the data coverage question. We have also covered this distinction at length in this letter, particularly in the *Novelty statement* at the beginning.

This is a much improved section, and addresses my concerns.

6. *'In summary, I think this is an interesting paper which has some promise. Before I can recommend its acceptance and publication in a journal such as Nature Communications, I would require that the questions I have raised, namely around comparisons to existing literature from the materials chemistry domain, and the applicability of the proposed method to more than 'high/low' fidelity classification and the requirements for data completeness in the low fidelity regimes, be addressed.'*

Response

We are thankful for the insightful questions and the appreciation of our work so far. These suggestions have unmistakably led to stronger results and a more general and capable framework. Naturally, we are happy and open to receiving additional feedback and continuing to refine our work.

References

- [Sch+17] Kristof Schutt et al. 'SchNet: A continuous-filter convolutional neural network for modeling quantum interactions'. In: *Advances in Neural Information Processing Systems*. Ed. by I. Guyon et al. Vol. 30. Curran Associates, Inc., 2017.
- [Gom+18] Rafael Gomez-Bombarelli et al. 'Automatic Chemical Design Using a Data-Driven Continuous Representation of Molecules'. In: *ACS Central Science* 4.2 (2018), pp. 268–276.
- [Sch+18] K. T. Schutt et al. 'SchNet – A deep learning architecture for molecules and materials'. In: *The Journal of Chemical Physics* 148.24 (2018), p. 241722.
- [GGG20] Johannes Gasteiger, Janek Groß and Stephan Gunnemann. 'Directional Message Passing for Molecular Graphs'. In: *International Conference on Learning Representations*. 2020.
- [Hu+20] Weihua Hu et al. 'Strategies for Pre-training Graph Neural Networks'. In: *International Conference on Learning Representations*. 2020.
- [Che+21] Chi Chen et al. 'Learning properties of ordered and disordered materials from multi-fidelity data'. In: *Nature Computational Science* 1.1 (Jan. 2021), pp. 46–53. ISSN: 2662-8457.
- [SUG21] Kristof Schutt, Oliver Unke and Michael Gastegger. 'Equivariant message passing for the prediction of tensorial properties and molecular spectra'. In: *Proceedings of the 38th International Conference on Machine Learning*. Ed. by Marina Meila and Tong Zhang. Vol. 139. Proceedings of Machine Learning Research. PMLR, July 2021, pp. 9377–9388.
- [But+22a] David Buterez et al. 'Graph Neural Networks with Adaptive Readouts'. In: *Advances in Neural Information Processing Systems*. Ed. by S. Koyejo et al. Vol. 35. Curran Associates, Inc., 2022, pp. 19746–19758.
- [But+22b] David Buterez et al. 'Multi-fidelity machine learning models for improved high-throughput screening predictions'. In: *ChemRxiv* (2022).
- [Ise+22] Clemens Isert et al. 'QMugs, quantum mechanical properties of drug-like molecules'. In: *Scientific Data* 9.1 (June 2022), p. 273. ISSN: 2052-4463.
- [Par22] Yang Jeong Park. *Edge Direction-invariant Graph Neural Networks for Molecular Dipole Moments Prediction*. 2022. arXiv: 2206.12867[cs.LG].
- [Wan+22] Yuyang Wang et al. 'Molecular contrastive learning of representations via graph neural networks'. In: *Nature Machine Intelligence* 4.3 (Mar. 2022), pp. 279–287. ISSN: 2522-5839.
- [But+23] David Buterez et al. 'Modelling local and general quantum mechanical properties with attention-based pooling'. In: *ChemRxiv* (2023).
- [Zha+23] Kaiwen Zha et al. *Supervised Contrastive Regression*. 2023. URL: https://openreview.net/forum?id=_QZlje4dZPu.

Reviewer #3 (Remarks to the Author):

Although authors made quite a lot of modifications to address my comments, I would still insist on my previous conclusion that the novelty of the manuscript is not large enough to enable the publication on this journal.

We would like to once again express our thanks to the Editor for considering our manuscript and to the Reviewers for their comprehensive and helpful insights. We are pleased that we have been able to resolve virtually all of the concerns presented during review, resulting in a stronger and more interesting overall package for the community.

Reviewer 2 has provided further commentary and clarifications for some of the points raised during peer review. We have found that they are, overall, in agreement with all of the revisions and updates to the manuscript. In the remarks, Reviewer 2 has left only a small number of actionable items. We have thus made some slight changes to our manuscript to incorporate them:

1. We have slightly lowered the claim of generality as this is best demonstrated through feedback and further advances from the community.
2. We have clarified the contributions of the related/existing work, in particular the use of Gaussian processes in the work of Fare et al. (2022).
3. We have added a few further citations in the relevant part of the *Discussion*, as suggested by the Reviewer.